# BEYOND ACCURACY: ARE TIME SERIES FOUNDATION MODELS WELL-CALIBRATED?

**Coen Adler[1] , Yuxin Chang[1] , Felix Draxler[1] , Samar Abdi[3] , Padhraic Smyth[1,2]**
[1]Department of Computer Science    [2]Department of Statistics
University of California, Irvine
[3]Google, Irvine
{ctadler,yuxinc20,fdraxler,smyth}@uci.edu
samaradbi@google.com

## ABSTRACT

The recent development of foundation models for time series data has generated considerable interest in using such models across a variety of applications. Although foundation models achieve state-of-the-art predictive performance, their calibration properties remain relatively underexplored, despite the fact that calibration can be critical for many practical applications. In this paper, we investigate the calibration-related properties of five recent time series foundation models and two competitive baselines. We perform a series of systematic evaluations assessing model calibration (i.e., over- or under-confidence), effects of varying prediction heads, and calibration under long-term autoregressive forecasting. We find that time series foundation models are consistently better calibrated than baseline models and tend not to be either systematically over- or under-confident, in contrast to the overconfidence often seen in other deep learning models.

## 1 INTRODUCTION

Prediction and modeling of time series data is ubiquitous in data analysis, with applications across a broad range of fields including climate science (Mudelsee, 2014), energy forecasting (Deb et al., 2017), healthcare (Crabtree et al., 1990), consumer behavior modeling (Goel et al., 2010), and financial forecasting (Tsay, 2005). Traditional statistical approaches such as linear autoregressive (AR) models and associated variants are well-established in the field (Hamilton, 1994; Hyndman & Athanasopoulos, 2018); they have been supplemented in recent years by a variety of machine learning approaches using richer model representations, including deep learning time series models such as N-BEATS (Oreshkin et al., 2020) and Informer (Zhou et al., 2021).

A more recent trend is the emergence of *time series foundation models (TSFMs)* (Nie et al., 2023; Liang et al., 2024). Unlike traditional statistical and machine learning approaches where models are fitted to time series from a single source, TSFMs are general-purpose models trained on a broad range of time series from various domains and are capable of zero-shot or few-shot forecasting on any time series in principle (Ye et al., 2024; Liang et al., 2024). This is appealing to practitioners in that only a single global model is required rather than retraining a new model for every time series (Bommasani et al., 2021; Benidis et al., 2022).

With this increase of interest in TSFMs, it becomes important to understand and characterize the calibration properties of such models. TSFMs, in general, produce conditional distributions over potential future values, rather than just single point forecasts (e.g., the expected value of the time series at a future time). This distributional information is important and useful in many applications. In particular, it can be essential for decision-making (Gneiting et al., 2007; Petropoulos et al., 2022), for example in downstream tasks such as anomaly detection (con, 2018) and in critical domains such as healthcare (Farah et al., 2014; Marlin et al., 2012). A natural question in this context is: **how well calibrated are TSFMs in terms of their conditional distributions, i.e., how well do the predicted probabilities match the observed data?**

While the calibration properties of classification and regression models have received considerable attention in machine learning in recent years (e.g., Guo et al. (2017); Song et al. (2019); Chung et al.

(2021)), the calibration properties of TSFMs remain relatively underexplored. To address this gap, we perform an empirical study of TSFMs and baselines with respect to *calibration-specific metrics*. Our systematic evaluation includes five state-of-the-art TSFMs and two well-established baseline methods in terms of their zero-shot forecasts across six univariate time series datasets with different temporal granularities. We measure calibration properties through a comprehensive set of metrics, such as Probabilistic Calibration Error (Dheur & Taieb, 2023; Kuleshov et al., 2018; Chung et al., 2021), in the context of a variety of different aspects of conditional uncertainty in model predictions.

In general, TSFMs produce conditional distributions using two approaches: some models predict the parameters of conditional density models (Rasul et al., 2023; Woo et al., 2024; Ansari et al., 2024a), while others predict sets of conditional quantiles (Das et al., 2024; Auer et al., 2025; Wang et al., 2025). To fully understand the calibration properties of TSFMs, we isolate the impact of different prediction heads by training individual quantile and distribution heads for each of the TSFM base architectures and then evaluate the effect of prediction heads on calibration performance.

Additionally, TSFMs vary in their approaches for long-term forecasting. For long-term forecasting beyond the *forecast horizon* (the prediction length of a single forward-pass), many TSFMs rely on autoregressive forecasting (Das et al., 2024; Ansari et al., 2024b; Cohen et al., 2025). Prior works (Auer et al., 2025; Wang et al., 2025) have shown that using autoregressive forecasting can have negative effects on model performance due to the reinitialization of the probabilistic forecast. Solutions include multi-patch forecasting (Woo et al., 2024) where models can predict sequential patches at a time, and stochastic autoregression which propagates probabilistic information to subsequent forecasts as described in Section 3.2. In this context, it is important to understand how limited forecast horizons can affect calibration on long-term forecasts. To investigate this, we compare TSFM calibration on long-term forecasting across varying forecast horizon lengths and autoregressive implementations.

The primary contributions of our work are as follows:

- We conduct a first-of-its-kind systematic calibration study with five state-of-the-art foundation models and two well-established baselines across six datasets from different domains[1].

- We use three calibration-specific metrics to quantify the overall calibration error and over- and under-confidence of prediction models; we analyze the calibration effects of various distributional and quantile prediction heads; and we investigate the effect of different long-range forecasting methods on calibration.

- We find that TSFMs are better calibrated than baseline models, tend to be neither systematically over- nor under-confident, are generally insensitive to the form of distributional prediction head, and consistently competitive in both prediction and calibration.

## 2 BACKGROUND AND RELATED WORK

### 2.1 PROBABILISTIC TIME SERIES FORECASTING

Given a *context* of $T$ observations $y_{1:T}$ for a time series, we evaluate the $L$-length forecasting performance on the prediction $y_{T+1:T+L} \mid y_{1:T}$. We denote the median prediction as $\hat{y}_t^{0.5}$ at each $t \in \{T+1, ..., T+L\}$ for point estimates, and predicted quantiles $\hat{y}_t^q$ to assess the uncertainty and calibration. The predicted quantiles are either produced directly by a multi-quantile prediction head for each of the $L$ future points, or can be computed from the predicted conditional density.

TSFMs can differ slightly in their approaches to data aggregation and functionality, but typically they tokenize time series in patches of a predetermined length (Nie et al., 2023). We denote the *forecast horizon* $H$ as the number of future time steps a model can forecast in a single forward-pass. A simple approach when the desired forecast length $L$ is greater than the model's forecast horizon $H$ is to use autoregressive (AR) forecasting based on point estimates, where the model will incorporate the previous forecasts $\hat{y}_{T+1:T+H}$ into the observed context. The model then uses this extended or shifted context $y_{1+H:T+H}$ to forecast the next set of patches $\hat{y}_{T+H+1:T+2H}$, recursively until the desired forecast $\hat{y}_{T+1:T+L}$ is reached (Ansari et al., 2024b; Cohen et al., 2025). Methods to

---

[1]Code available at `https://github.com/Coaster41/Beyond-Accuracy-TSFM-Calibration`

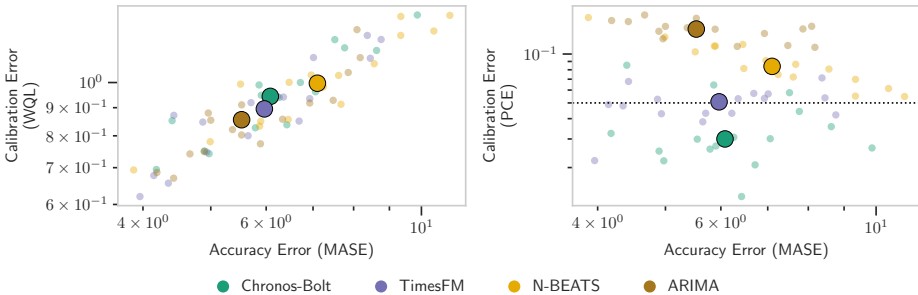

Figure 1: **Weighted Quantile Loss (WQL) calibration metric can be highly correlated with point accuracy error (MASE).** Results shown for WQL (left) and PCE (right) plotted against MASE on the Glucose dataset used in our experiments. Background markers are results for each model and each individual time series; larger centroids are medians across time series. **On this dataset, WQL incorrectly identifies ARIMA as the best calibrated model.**

reduce the reliance on AR forecasting include extending the number of simultaneous output patches in multi-patch forecasting (Woo et al., 2024), increasing output patch length (Das et al., 2024), and incorporating missing-data masks to extend the context into the future (Auer et al., 2025). We empirically compare these approaches for long-term forecasting in Section 4.4.

In TSFMs, patched time series are passed into a model-specific backbone, which translates the embeddings into a latent space as an input to the final projection block. These probabilistic predictions can then be directly predicted as quantiles through a multi-headed projection block or a model can use the latent activations to output the parameters to a distribution where the full densities can be recovered. Some prediction heads can better represent different types of data, for instance, a negative binomial would be well-suited for count data, while a Student's $t$ distribution would be appropriate for a general continuous-valued time series. Furthermore, parameterizations of mixture of distributions, as well as quantile heads, have also been proposed (Woo et al., 2024; Das et al., 2024; Cohen et al., 2025) to allow for additional flexibility in terms of how TSFMs can model conditional distributions. We evaluate these different projection blocks across a range of domains to identify if certain heads are better calibrated for specific datasets in Section 4.3.

## 2.2 RELATED WORK ON CALIBRATION

Predictive calibration has long been recognized as a topic of importance in statistics and machine learning (Brier, 1950; Gneiting et al., 2007). In recent years, a number of different studies have investigated the calibration of deep learning models, and have shown that such models are often systematically susceptible to overconfidence, for example in image classification with deep models (Guo et al., 2017; Ye et al., 2023; Pinto et al., 2022) and in multiple question-answering tasks for large language models (Jiang et al., 2021; Mielke et al., 2022; Xiong et al., 2024).

In the evaluation of time series forecasting models, calibration is also an important component (Makridakis et al., 2022; Aksu et al., 2024; Das et al., 2024). One of the most commonly used calibration metrics is the Continuous Ranked Probability Score (CRPS), which directly compares the cumulative distribution function (CDF) of model predictions to the observed value's CDF (Hersbach, 2000). Since computing CRPS can be intractable and some TSFMs only produce quantiles rather than full conditional densities, the Weighted Quantile Loss (WQL) is also often used as a discrete surrogate for CRPS (Woo et al., 2024; Aksu et al., 2024), defined as the pinball (or quantile) loss scaled by the absolute sum of the true values. Another common calibration metric is Mean Scaled Interval Score (MSIS) (Makridakis et al., 2020), which takes the mean difference in upper and lower bound predictions and adds an error penalty when the true value lies outside the bounds.

Using these metrics, prior works have claimed that TSFMs are better calibrated than baseline methods like ARIMA and N-BEATS (Aksu et al., 2024; Ansari et al., 2024a; Auer et al., 2025). However, Chung et al. (2021) proved that the metrics above (CRPS, WQL, MSIS) measure a combination of both probabilistic calibration and sharpness (Gneiting et al., 2007), rather than calibration alone. This combination can result in an imbalance in evaluation that can skew towards prioritizing predic-

tive sharpness (Chung et al., 2021). Figure 1 shows a concrete example of how PCE and WQL can differ in practice: WQL can be highly correlated with Mean Absolute Scaled Error (MASE), while PCE directly measures calibration. In the next section, we discuss calibration-focused metrics that address this issue by focusing solely on measuring calibration.

# 3 EXPERIMENTAL SETUP

## 3.1 METRICS

The first calibration metric we use is **Probabilistic Calibration Error (PCE)** (Kuleshov et al., 2018; Dheur & Taieb, 2023):

$$\text{PCE} = \frac{1}{|Q|} \sum_{q \in Q} \left| q - \frac{1}{L} \sum_{t=T+1}^{T+L} \mathbf{1}[y_t \leq \hat{y}_t^q] \right|, \tag{1}$$

where $q \in Q = \{0.1, 0.2, ..., 0.9\}$ is the set of quantiles that we average over in our experiments. Intuitively, PCE measures the differences between empirical and predicted CDFs with lower values indicating better-calibrated models. PCE is lower-bounded by 0 and upper-bounded by 0.5 for the case where predicted quantiles are always all above or below the observed value $y_t$.

However, well-calibrated models are not sufficient to produce useful forecasts: a model could for example always predict the marginal distribution, independent of the inputs. In this context, a metric that specifically captures *sharpness* (the concentration of the predictive distributions) is also important in an overall evaluation of calibration (Gneiting et al., 2007; Kuleshov et al., 2018). A simple surrogate for sharpness is the width of a predicted confidence interval, e.g., where an 80%-confidence interval is the interval between the symmetric $q_{\text{low}} = 10\%$ and $q_{\text{high}} = 90\%$ quantile predictions. We refer to this as **Scaled Interval Width (SIW)** where $s$ is the confidence associated with the interval (i.e., $s = q_{\text{high}} - q_{\text{low}} = 80\%$ in the preceding example):

$$\text{SIW}_s = \frac{1}{L} \sum_{t=T+1}^{T+L} \frac{\hat{y}_t^{q_{\text{high}}} - \hat{y}_t^{q_{\text{low}}}}{y^{q_{\text{high}}} - y^{q_{\text{low}}}}. \tag{2}$$

Models with lower SIW values (i.e., tighter prediction intervals) are more confident in their predictions, while models with larger SIW values (i.e., wider-spread predictions) are less confident.

We are also interested in whether a model is systematically miscalibrated in one direction or another. To quantify this, we define the metric **Centered Calibration Error (CCE)** that compares the amount of observed data in a predicted interval with the associated confidence $s$:

$$\text{CCE} = \frac{1}{|S|} \sum_{s \in S} s - \frac{1}{L} \sum_{t=T+1}^{T+L} \mathbf{1}\left[\hat{y}_t^{q_{\text{low}}} \leq y_t \leq \hat{y}_t^{q_{\text{high}}}\right]. \tag{3}$$

A model's over- or under-confidence can be identified by combining the direction of CCE and its SIW. Positive CCE values indicate there is more observed data outside the predicted interval than expected by the confidence level; together with a low SIW value, we can infer that a model is overconfident. On the other hand, negative CCE values and larger SIW values imply that the model is under-confident. For both CCE and SIW, we average over $s \in \{0.2, 0.4, 0.6, 0.8\}$ in our analyses.

Finally, to assess the point accuracy of TSFMs (independently from any calibration error) we use **Mean Absolute Scaled Error (MASE)** for the predicted median, scaling the Mean Absolute Error (MAE) by a naive predictor (Hyndman & Athanasopoulos, 2018):

$$\text{MASE} = \frac{\frac{1}{L} \sum_{t=T+1}^{T+L} |\hat{y}_t^{0.5} - y_t|}{\frac{1}{L-1} \sum_{t=T+2}^{T+L} |y_t - y_{t-1}|}. \tag{4}$$

Note that for multi-step predictions ($L > 1$) that the MAE of the naive predictor in the denominator is in effect using information "from the future," with the result that the overall MASE may take values greater than 1 even for useful models. The absolute values of MASE are not important, but the relative MASE values across different predictors $\hat{y}$ are what we will focus on.

## 3.2 MODELS

**Base Models**  We evaluate calibration properties of five TSFMs in terms of their zero-shot forecasts: Chronos-Bolt (Ansari et al., 2024a;b), TimesFM (Das et al., 2024), Moirai 2.0 (Woo et al., 2024; Taha Aksu et al., 2025), TiRex (Auer et al., 2025), and YingLong (Wang et al., 2025). As baselines, we use ARIMA (Hyndman & Khandakar, 2008) and N-BEATS (Oreshkin et al., 2020) to represent well-known parametric and neural time-series prediction alternatives to TSFMs. All selected TSFMs have been pretrained on various large pretraining datasets and use a quantile prediction head. Additionally, they all use transformers, except for TiRex which uses an xLSTM (Beck et al., 2024) backbone. TimesFM, Moirai 2.0, and TiRex use a decoder-only architecture with a casual attention mechanism, while Chronos-Bolt uses the encoder-decoder backbone from the T5 model (Raffel et al., 2020) and YingLong uses a bi-directional encoder-only architecture.

**Autoregressive Methods**  Autoregressive (AR) forecasting is often necessary for models with a limited forecast horizon for long-term forecasting. Each model implements AR forecasting slightly differently. Chronos-Bolt and TimesFM use a naive point-based AR, where only the mean or median predictions are autoregressively added to the context. This method while simple is known to significantly harm probabilistic forecasting due to the re-initialization of the context each iteration (Auer et al., 2025). Improvements to the naive method typically require forecasting each time step multiple times with different AR contexts to propagate probabilistic information across forecast iterations: Moirai 2.0 implements a branching approach which autoregressively forecasts using a separate context for each quantile. The Toto TSFM (Cohen et al., 2025) uses a trajectory approach which produces $n$ (usually $\gg |Q|$) independent AR forecasts or trajectories, where each trajectory is produced similarly to the point-based approach. Instead of adding the mean or median forecast to the context, it adds a random sample from the predicted distribution at each time step. We explain these methods in more detail in Appendix A.3. These AR methods trade off robustness for computational efficiency, where the TimesFM and Chronos-Bolt approach only forecasts each time step once, while the branching (Moirai 2.0) and trajectory approach (Toto) require $|Q|$ and $n$ forecasts per time step respectively. In addition to comparing these AR methods, we evaluate how larger or smaller forecast horizons affect calibration on long-term AR forecasting.

## 3.3 DATASETS

We selected evaluation datasets representing a range of tasks differing in time-step granularity, seasonality, and forecasting difficulty across a variety of domains. To the best of our knowledge, these datasets were not used in the training of the foundation models (except the M5 data for the Moirai 2.0 and YingLong models). Each dataset is split into a train and test set where the non-TSFMs, ARIMA and N-BEATS, are trained independently on the train set, where hyperparameters are chosen by AutoARIMA and grid search respectively. All models are evaluated on the held-out test set. Table 1 summarizes the dataset statistics. Our results on each dataset are aggregated over multiple settings (see Section 3.1), prediction steps, and all possible context-prediction combinations in the test set for each time series, i.e., effectively generating many more time series forecasts than the number of time series (see the last column in Table 1). Additional details are provided in Appendix A.2.

Table 1: Datasets used in calibration experiments.

|  | # Time Series | Granularity | Time Steps per Series | # Total Forecasts |
|---|---|---|---|---|
| Reviews | 239 | Hourly | 13,000 | 360,490 |
| Shopping (M5) | 70 | Daily | 1,912 | 62,160 |
| Glucose | 16 | 5 Mins | 1,686 | 13,840 |
| Heart-Rate | 6 | Second | 744 | 1,800 |
| Crime | 5 | Daily | 6574 | 4,110 |
| Patents | 83 | Monthly | 408 | 10,956 |

To evaluate the effect of different prediction heads for the TSFMs, we trained prediction heads using a large diverse dataset that is comparable to those used in TSFM pretraining. Specifically, we selected **TSMixup** from  Ansari et al. (2024a) as it is independent of the six datasets we use to evaluate

the models. This allows the trained projection blocks to be an equivalent plug-in replacement to the default quantile projection heads of the selected models.

## 4 EXPERIMENTS

Based on the models, datasets, and metrics outlined above, we found that foundation models exhibit competitive and often better point-forecasting performance compared to baselines, with the TSFMs often having a lower MASE than N-BEATS and ARIMA (see Figure 2). The Glucose and Patents datasets have significantly higher MASE numbers than the other datasets—this may be due to the fact that for each of these datasets there appears to be little significant linear dependence beyond 1 or 2 lags (see partial autocorrelation plots in Figure 6 in Appendix B.4).

TSFMs also exhibit significantly better calibration performance than baseline models. We discuss the main results on calibration in detail below. In the Appendix, we also include additional figures and findings, where we discuss the empirical correlation of WQL, MSIS, and MASE, along with details on the calibration of tail probabilities and additional prediction heads and datasets.

### 4.1 ARE FOUNDATION MODELS WELL-CALIBRATED?

Previous works have indicated that TSFMs are well calibrated using CRPS, WQL, and MSIS (Aksu et al., 2024; Auer et al., 2025), but as discussed in Section 3.1 and in Chung et al. (2021), these metrics can be biased towards sharpness and accuracy rather than reflecting calibration per se. In our experiments, we evaluate directly if TSFMs are well-calibrated, focusing on the PCE metric, and **we find that, yes, TSFMs are generally better calibrated than the baseline models of N-BEATS and ARIMA**: see top plot in Figure 3. For a general sense of scale, PCE values below 0.05 or 5% error relative to the quantiles can loosely be considered to be much better calibrated than values larger than say 0.15, in the context of the range of PCE being between 0 and 0.5. The TSFMs are in general close to or below 5% in PCE error, except for the relatively difficult Patents dataset where all methods (TSFMs and baselines) have high calibration error. No single TSFM significantly dominates in calibration performance over the others.

A natural additional question is how calibration performance is affected by how far in time a model prediction is from the context. Naturally, as both TSFMs and baselines predict further into the future, both the point accuracy (MASE) and calibration error (PCE) degrade (see Figures 7 and 8 in the Appendix). However, calibration overall remains relatively stable with TSFMs achieving PCE values close to or below 5% even 64 time-steps out (e.g., for Reviews, M5, and Crime data), unlike the baseline models which have a consistently high PCE over all prediction lengths.

As a control experiment, we also evaluated the calibration performance of TSFMs on two synthetic datasets, one with pure IID noise $y_t = \epsilon$ and the other a noisy first-order linear process $y_t = \alpha y_{t-1} + (1 - \alpha)\epsilon$, with $\epsilon \sim \mathcal{N}(0, 1)$ and $\alpha = 0.9$. For these datasets the question of interest is whether TSFMs might overfit (for either calibration or point prediction) relative to a baseline like ARIMA (which in principle can be perfect on these datasets). As shown in the lower right two plots in Figure 9 in the Appendix, the TSFMs do not overfit (they perform well on both MASE and PCE). ARIMA does well on MASE but is not as well-calibrated (PCE) (nor is N-BEATS) as the TSFMs.

In summary, while past works have shown that TSFMs perform well on WQL and CRPS metrics (e.g., Aksu et al. (2024); Ansari et al. (2024a); Auer et al. (2025)) as we discussed earlier (see Figure 1 and further in Appendix B.2), WQL alone is not necessarily a reliable indicator of calibration performance. Our results provide direct confirmation that TSFMs tend to be well-calibrated.

### 4.2 ARE FOUNDATION MODELS SYSTEMATICALLY BIASED IN MISCALIBRATION?

For image and text modalities, deep models are well-known to often be overconfident (Ye et al., 2023; Xiong et al., 2024; Kapoor et al., 2024). In the context of time series, it is natural to ask if TSFMs similarly exhibit systematic biases, either towards over- or under-confidence.

To answer this, we evaluated the CCE metric across datasets and models. The lower plot in Figure 3 shows the directionality of model confidence. Except for the Patents dataset, where all models are overconfident, we find that **TSFMs tend not to be systematically over- or under-confident**. In

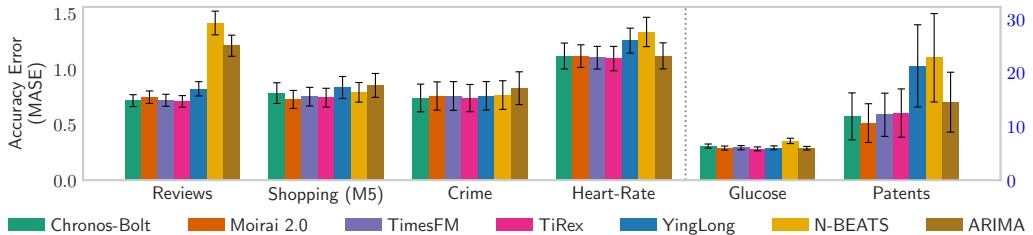

Figure 2: **TSFMs offer competitive point forecasting performance compared to the baseline models, often having better accuracy than the baselines.** $y$-axis: Mean Absolute Scaled Error (MASE) measuring point accuracy error of the median prediction (lower is better), Glucose and Patents use their own $y$-axis scale (on the right). The error bars are computed as the Standard Error of the Mean (SEM) across timeseries.

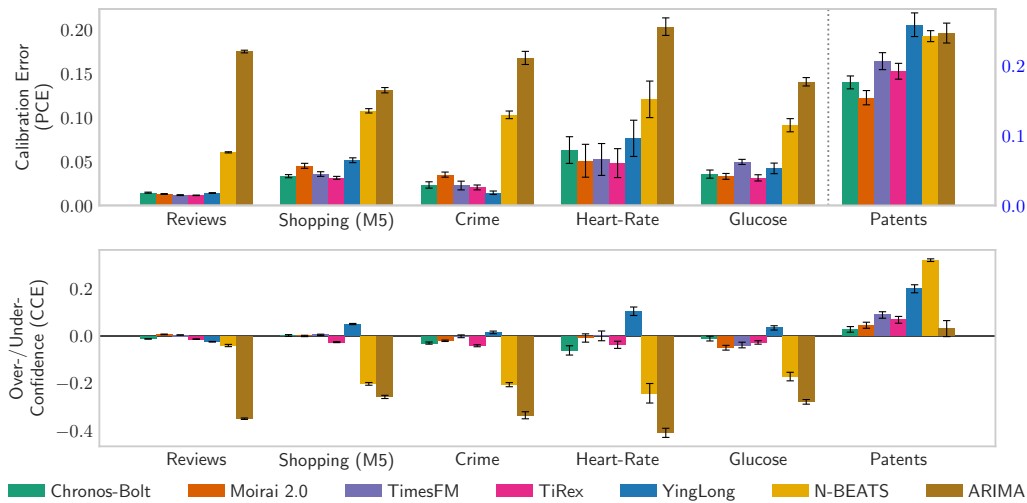

Figure 3: **TSFMs are better calibrated than the baseline models and are not systematically over- or under-confident.** Top: Probabilistic Calibration Error (PCE) across datasets and models using the default quantile projection block. (Patents PCE uses its own $y$-axis scale (on the right)). Lower PCE values are better. Bottom: Centered Calibration Error (CCE) evaluating systematic overconfidence (positive) or under-confidence (negative).

contrast, N-BEATS and ARIMA are consistently under-confident As shown in Figure 12 in the Appendix, the interval width (SIW) and centered calibration (CCE) tend to have a negative correlation: models with wider confidence intervals tend to have a smaller CCE or are under-confident.

It is noteworthy that TSFMs are not systematically overconfident, in the way that foundation models from the text and image domains tend to be. This can likely be explained by the fact that TSFMs are being directly trained with a calibration-aware loss (i.e., trained to minimize WQL), while text and image models are trained to minimize reconstruction or classification error.

## 4.3 How Do Prediction Heads Affect Calibration?

A major design choice of TSFMs is selecting a suitable prediction head. Multiple approaches have been presented in the literature, including quantile prediction heads (the default choice for all TSFMs in our experiments), parametric distribution heads (e.g., Lag-Llama (Rasul et al., 2023) with Student's $t$), categorical distribution heads (e.g., Chronos (Ansari et al., 2024a)), and semi-parametric

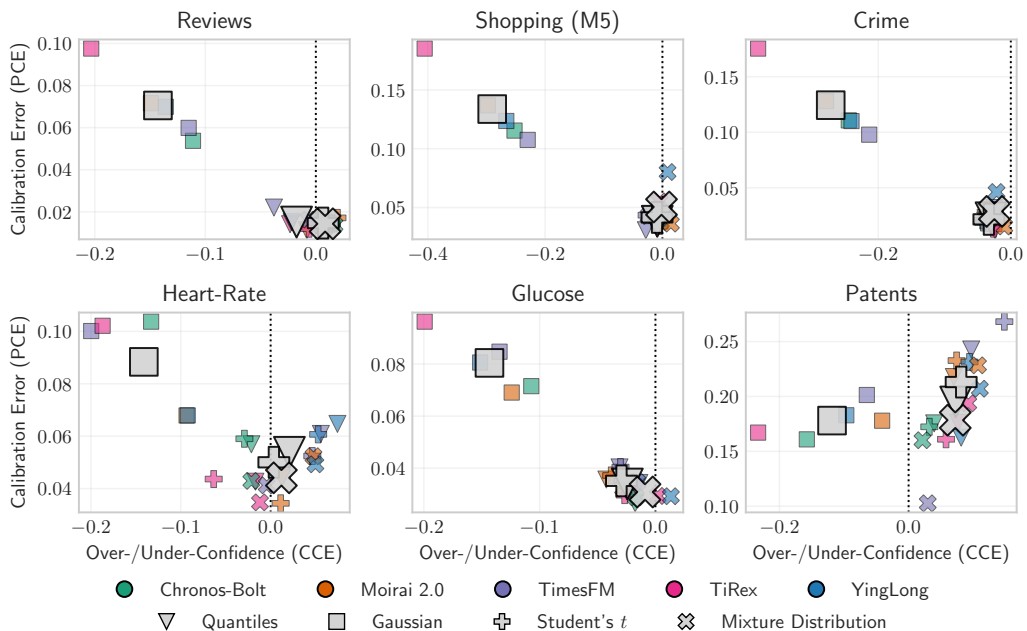

Figure 4: **Gaussian prediction heads are under-confident and are in most cases outperformed by the other prediction heads.** Calibration properties of various prediction heads with Centered Calibration Error (CCE) on the $x$-axis and Probabilistic Calibration Error (PCE) on the $y$-axis. Large positive CCE values indicate overconfident predictions, while large negative CCE values show under-confidence. Background markers are the results for each head (shape) with each backbone model (color), while the larger gray centroids are the mean results across models.

mixture heads such as Toto (Cohen et al., 2025) and Moirai 1.0 (Woo et al., 2024) which use a variety of components such as Gaussian, Student's $t$, log-normal, and negative-binomial distributions.

An important question is whether the form of a prediction head for a model affects its calibration performance? To address this question, we used the latent information produced by each pretrained TSFM backbone and trained different prediction heads (using TSMixup) for each, specifically: a Gaussian distribution, a Student's $t$ distribution, and a mixture distribution containing a Gaussian, Student's $t$, log-normal, and a Laplace distribution. We also retrained a separate quantile projection head on TSMixup as a control to compare against and found that the trained projection heads correctly learn the latent information of the original foundation model with the trained quantile heads having equivalent forecasting performance as the original quantile head.

We then evaluated all combinations of models and prediction heads using the PCE and CCE metrics. As shown in Figure 4, the **quantile, Student's $t$, and mixture distribution heads have very similar calibration error across all datasets**, indicating that there is no significant advantage for any of the three over the others. On the other hand, **the Gaussian distribution's calibration results are significantly worse** than the other heads. In particular, the Gaussian heads are consistently under-confident with CCE scores ($x$-axis of Figure 4), always lower than the other heads. This under-confidence also translated to consistently higher calibration error. We speculate that the limited expressiveness of the Gaussian distribution results in poor calibration while the more expressive distributions are better calibrated without suffering from overfitting issues.

### 4.4 How Do AR Methods Affect Calibration in Long-Term Forecasting?

It is well known that autoregressive (AR) models can deteriorate when making long-term predictions when the desired forecast length $L$ is much longer than the model's forecast horizon $H$ (Auer et al., 2025). Different approaches to AR forecasting have been proposed, with different tradeoffs in terms of how errors accumulate and how much computational effort is involved. An important question from a calibration perspective is how do the different AR forecasting methods affect calibration

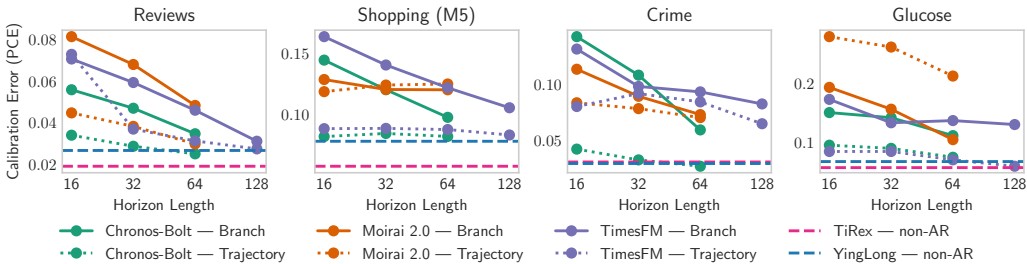

Figure 5: **For long-term AR forecasting, increasing the horizon length and using the trajectory AR method produces better calibrated forecasts.** The figure compares Probabilistic Calibration Error (PCE) for long-term forecasting using autoregression on the $y$-axis with AR prediction horizon on the $x$-axis. The color of the line depicts the model used and the line style indicates the AR method. The pink and blue dashed horizontal lines are the reference PCE for TiRex and Yinglong which are capable of long-term forecasting without the use of AR.

performance? To investigate this, we evaluated how calibration properties of TSFMs vary as a function of horizon length $H$ and AR method. We focused primarily on the Chronos-Bolt, TimesFM, and Moirai 2.0 models in these experiments given that TiRex and YingLong model architectures allow for long-term forecasting natively without requiring AR.

For both the trajectory and branching AR methods, the **predictions from models with a shorter forecast horizon $H$ have poorer calibration** on a fixed forecast length $L$. This is more pronounced in the branching method, where forecasts with horizon lengths of 16 have notably worse PCE than with 64 or 128 as shown in Figure 5. The trajectory AR approach has generally lower PCE than the branching method when comparing with the same forecast horizon. The explanation for this becomes more clear when viewing CCE from Figure 17 in the Appendix with respect to these methods. **All autoregressive TSFMs are consistently overconfident in long-term forecasting**. While both methods show that shorter forecast horizons results in more confident forecasts, the CCE values for the branching method are often greater than 0.15 for horizon lengths of 16 and 32. This overconfidence reduces steeply as the horizon length increases thus reducing overall calibration error. Although the trajectory method can be more computationally expensive than the branching approach, our results indicate that the trajectory method is better calibrated for long-term forecasting. The non-AR approaches of TiRex and YingLong are significantly more efficient than both AR methods, and they tend to be better calibrated and were not significantly over- or under-confident. Future work in long-term forecasting should prioritize models with longer forecast horizons and alternatives to AR-based approaches.

## 5 CONCLUSION

Understanding the calibration properties of TSFMs closes a gap in the literature, which has primarily focused to date on point accuracy. Calibration is crucial, however, to accurately quantify the inherent uncertainties associated with time-series forecasting.

In this paper, we evaluated the calibration properties of current leading TSFMs and found that they are consistently well-calibrated relative to the N-BEATS and ARIMA baselines. Unlike the baselines, which are under-confident, the TSFMs show no signs of systematic over- or under-confidence in short-term forecasting. When replacing the projection heads of the TSFMs, the Gaussian prediction head results in consistently under-confident forecasts across the different model backbones, while other distribution and quantile heads are all well-calibrated without significant differences. For long-term forecasting using autoregression, having larger prediction horizons and using the trajectory AR approach over the branching method decrease calibration error and reduces overconfidence.

A limitation of our study is that our evaluations only considered TSFMs for zero-shot univariate time series and calibration relative to a fixed set of quantiles $q \in \{0.1, \ldots, 0.9\}$. As such, worthwhile extensions of our work could be to investigate calibration in the context of fine-tuning, to extend multivariate data, and to examine higher-resolution quantiles. Another important practical

direction for future investigation is how distribution shift and non-stationarity may affect the calibration performance of both TSFMs and baselines, given the well-known sensitivity of the calibration performance of deep classification models to distribution shift (Snoek et al., 2019).

ACKNOWLEDGMENTS

This work was supported in part by the Hasso Plattner Institute (HPI) Research Center in Machine Learning and Data Science at the University of California, Irvine, by a Google faculty award, by the National Science Foundation under award NSF RISE-2425932, and by the National Institutes of Health under awards 1R01CA297869-01 and R01-LM013344. FD acknowledges funding from the National Science Foundation (NSF) through an NSF CAREER Award IIS-2047418, IIS-2007719, the NSF LEAP Center, and the Chan Zuckerberg Initiative.

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

# A APPENDIX

## A.1 LLM USAGE STATEMENT

LLMs were used by the authors as a tool to assist in producing code for experiments. LLMs were not used to aid in the writing of the paper or during research ideation.

## A.2 ADDITIONAL DETAILS ON EXPERIMENTAL SETUP

We report the default experiment parameters used in the main paper results in the following table, where the train/test split size varies by dataset, with train sets containing approximately 700 time steps per time series:

Table 2: Experimental setup details per dataset.

| | Prediction Length ($L$) | Context Size ($T$) | Seasonality (ARIMA only) | Train/test split |
|---|---|---|---|---|
| Reviews | 64 | 512 | 24 | 2020-01-31 23:00 |
| Shopping (M5) | 64 | 512 | 7 | 2015-04-23 |
| Glucose | 64 | 512 | 1 | 2020-02-16 12:43 |
| Heart-Rate | 64 | 128 | 1 | 2000-01-01 00:04:59 |
| Crime | 64 | 512 | 7 | 2015-01-01 |
| Patents | 64 | 128 | 1 | 2004-01-01 |

Prediction length is fixed at 64 time steps across all datasets for short-term forecasting experiments while long-term forecasting experiments used a desired forecast length of 256 time steps.

Baseline models (N-BEATS and ARIMA) use the training data (dates before train/test split column) for parameter selection and training. The train/test split was determined based on the minimal size of the train set for training N-BEATS and ARIMA models, limited by time series length. Models forecast the first 64 time steps of the test set using the end of the training set as the context. The forecast date and non-AR context are then shifted by the stride, forecasting an additional 64 time-steps. The models forecast repeatedly until the end of the time series. For a stride $d \in \{1, 4, 8\}$ dependent on time series length, the first forecast predicts $\hat{y}_{T+1:T+L}|y_{1:T}$ while the second forecast is shifted by the stride to $\hat{y}_{T+1+d:T+L+d}|y_{1+d:T+d}$.

Seasonality is a hyperparameter for the ARIMA models, used for initialization. We limit the max lag (context) for the ARIMA model to 64 for all datasets. We train an ARIMA model with AutoARIMA on the train set and then use the selected hyperparameters to train the models for evaluation. We train a new ARIMA model each time we shift the context.

For N-BEATS we perform a grid search on a variety of hyperparameters (see Table 3) using the training data and select the model with the lowest WQL on the same data. Quantile loss outperforms normal distribution loss on all datasets. When forecasting with distribution loss in N-BEATS, we use the default 100 samples. We train a single N-BEATS model on each dataset, training across multiple time series and reuse the same learned parameters each time we shift the context.

Table 3: N-BEATS hyperparameter grid search space.

| Hyperparameter | Search Space |
|---|---|
| Epochs | {100, 1000} |
| Learning Rate | {0.001, 0.0001} |
| Early Stop Patience Steps | {-1, 2} |
| Number of Blocks | {(1,1,1), (3,3,3)} |
| Validation Check Steps | {10, 100} |
| Loss Function | {Normal Distribution Loss, Quantile Loss} |

We describe the AR methods in depth in the next section and used $n = 100$ as the number of trajectories for the trajectory AR method. When sampling from the quantile heads in the trajec-

tory method, we linearly interpolate between the quantiles to convert the quantile heads into to a continuous distribution.

### A.3 AUTOREGRESSIVE METHODS

The **naive** AR method used by TimesFM and Chronos-Bolt adds only the median or mean forecasts to the context. For example if the desired forecast length is $L = 256$ and the model's forecast horizon is $H = 64$, the model will generate a probabilistic forecast $\hat{y}_{T+1:T+64}$ given the observed context $y_{1:T}$. To make the next 64-step forecast, it adds the mean or median forecast, derived from $\hat{y}_{T+1:T+64}$, to the original context. The result is a shifted context $y_{65:T+64}$ containing a mix of observed values and forecasted values. The model can then use the shifted context to produce the forecasts for $\hat{y}_{T+65:T+128}$. This process is repeated until all 256 probabilistic forecasts have been completed, and the quantiles can be obtained directly from the probabilistic forecasts.

The **branching** approach from Moirai 2.0 is different in that it requires the model to forecast the same time steps multiple times. The first 64 time steps are normal, relying only on the observed contexts and producing $|Q|$ quantile forecasts $\hat{y}_{T+1:T+64}^q$, where $q \in Q$. The context is duplicated or branched into $|Q|$ new extended contexts, where each context is extended by the forecasts at a specific quantile. The next 64 steps $\hat{y}_{T+65:T+128}^q$ is forecasted using all $|Q|$ contexts, which requires running the model $|Q|$ times independently, once for each of the $|q|$ contexts. Because the model has been run $|Q|$ times and each time the model produces $|Q|$ quantiles, there will be a total of $|Q|^2$ forecasts for each time step. Rather than increasing the number of contexts exponentially from $|Q|$ to $|Q|^2$, the $|Q|^2$ quantile forecasts at each time step are aggregated back down to a set of $|Q|$ quantile predictions. This is done by taking the $Q$ quantile of the $|Q|^2$ forecasts (for each time step). These aggregated quantiles become the returned quantile forecasts and are used to extend and shift the corresponding $|Q|$ contexts.

The Toto TSFM (Cohen et al., 2025) uses a **trajectory**-based AR method where the model generates $n$ independent AR forecasts or trajectories. Each trajectory is produced similarly to the naive method where a single point forecast (at each time step in $H$) is added to the context for AR. However, instead of using the mean or median, the model samples from the predicted density and uses the samples as the point forecast (one sample for each time step). The model will create $n$ independent trajectories, each with separate contexts. The final quantile forecasts are produced by taking the quantiles of the $n$ trajectories at each time step independently.

### A.4 MODELS

We primarily selected models with diverse backbones that were state-of-the-art on the GIFT-EVAL benchmark as of the time of writing (Aksu et al., 2024) and pre-trained on time series datasets.

**TimesFM**  TimesFM (Das et al., 2024) uses a decoder-only stacked transformer architecture, and provides probabilistic predictions at pre-trained fixed quantile heads. The training objective combines mean squared error (MSE) and quantile loss Hyndman & Athanasopoulos (2018); Gneiting et al. (2007). The model is trained with larger output patches than inputs, and thus able to make joint predictions of forecast quantiles over lengths $H \leq 128$. We use the `timesfm-2.0-500m-pytorch` version for our experiments. [2]

**Moirai 2.0**  Moirai 2.0 (Taha Aksu et al., 2025) is a recent update on Moirai 1.0 (Woo et al., 2024) transitioning from an encoder-based transformer that uses a mixture distribution projection head to a decoder-only backbone with a quantile head. Moirai 2.0 is implemented using AR with multi-patch forecasting support. In our experiments, we used a patch size of 16 and set the model to jointly predict 4 patches at a time for a max prediction horizon of $H = 64$. We use the `moirai-2.0-R-small` version for our experiments. [3]

---

[2]TimesFM checkpoint on Hugging Face: `https://huggingface.co/google/timesfm-2.0-500m-pytorch`.

[3]Moirai 2.0 checkpoint on Hugging Face: `https://huggingface.co/Salesforce/moirai-2.0-R-small`

**Chronos-Bolt**    Chronos-Bolt (Ansari et al., 2024b) is a foundation model with an backbone from the T5 family of encoder-decoder language models   Raffel et al. (2020).   Real-valued time series is tokenized into fixed vocabularies via scaling and quantization.   Unlike its predecessor Chronos (Ansari et al., 2024a) which used a categorical prediction head forecasting the tokenized patches within a set vocabulary, Chronos-Bolt directly predicts quantiles using an output patch length of 64. We use the `chronos-bolt-base` version for our experiments. [4]

**TiRex**    TiRex (Auer et al., 2025) deviates from most TSFMs by using an xLSTM (Beck et al., 2024) decoder-only backbone. They apply contiguous patch masking during pretraining to allow for long-term forecasting without the need for AR. For long-term forecasts at inference time, TiRex pads and shifts the context so that the model does not need to condition on previous forecasts. Their implementation allows for multi-patch accelerated roll-out by using the forecasts at multiple patches in a single forward pass, rather than only using a single patch before padding the context. We use the base `TiRex` version for our experiments and set the accelerated roll-out to 4 patches with patch size of 32. [5]

**YingLong**    YingLong (Wang et al., 2025) employs a bi-directional U-net encoder-only transformer backbone. This enables the model to use delayed chain-of-thought reasoning by forecasting beyond the desired forecast length. The desired forecast can then condition on both the observed context and the future delayed chain-of-thought for more accurate predictions. We use the `YingLong_110m` version for our experiments and use an extended forecast horizon of 2048. [6]

**N-BEATS**    N-BEATS (Oreshkin et al., 2020) is a deep neural architecture that has been designed for the purposes of time series predictions.  Similar to TimesFM, N-BEATS jointly forecasts the entire prediction horizon in a single forward pass.

**ARIMA**    We use Nixtla's `StatsForecast` implementation of AutoARIMA (Hyndman & Khandakar, 2008) to automatically select the optimal ARIMA parameters for each time series on the training set.  The model is then refit on all earlier data before each forecast on the evaluation set.  The ARIMA implementation uses Kalman filters to recursively predict the mean and variance. Quantiles are computed by fitting a normal distribution to the forecasts and using the inverse CDF (PPF) with the appropriate $z$-scores.

## A.5    Prediction Heads

To evaluate the impact the prediction head has on calibration, we replaced the default quantile projection heads of each model with a fine-tuned version of each head we tested. To train each head, we cached the embeddings outputted by each model's backbones and used them as the input to each of the heads. Specifically, we trained an independent quantile, Gaussian, Student's $t$, and mixture distribution head.  For the mixture head we replicate the Moirai 1.0 (Woo et al., 2024) prediction head using a Gaussian, Student's $t$, log-normal, and replace Negative-Binomial with the Laplace distribution due to issues related to model convergence during training.

## A.6    Datasets

We evaluate models on three human behavior datasets: (i) a **Reviews** dataset consisting of hourly counts of Amazon product reviews (Hou et al., 2024) and Google Places reviews (Li et al., 2022), (ii) a modified **Shopping (M5)** dataset (Makridakis et al., 2022) consisting of the daily number of products being sold at different locations, and (iii) an NYC **Crime** report dataset (New York City Police Department, 2025) aggregating daily crime occurrences.

For the **Reviews** ensemble dataset, we aggregated Amazon Hou et al. (2024) and Google Li et al. (2022) reviews by product and location category and sampled 239 of the most abundant categories

---

[4]Chronos-Bolt checkpoint on Hugging Face: `https://huggingface.co/amazon/chronos-bolt-base`

[5]TiRex checkpoint on Hugging Face: `https://huggingface.co/NX-AI/TiRex`

[6]YingLong checkpoint on Hugging Face: `https://huggingface.co/qcw2333/YingLong_110m`

from the ensemble. We binned the event data as count data at an hourly granularity and trimmed the dataset to the most active time range from 2020-01-01 08:00:00 to 2021-06-25 23:00.

We aggregated the daily **Shopping (M5)** Makridakis et al. (2022) dataset to reduce sparsity by binning each product by their product department and store ID, totaling 70 time series.

We aggregated the NYC **Crime** reports dataset New York City Police Department (2025) by number of daily reports and cutoff the dataset to only include reports between the years 2006 and 2023. We split the dataset into time series based on borough.

Many datasets used in training TSFMs are related to human behavior and natural phenomena that exhibit periodic trends (e.g., 24-hour effects). As noted in Gu et al. (2025), the strong predictive accuracy of foundation models on many of the datasets used in machine learning evaluations does not necessarily translate into high predictive accuracy in applications such as vital sign forecasting in healthcare. To analyze model performance on datasets that differ in this respect from the pretraining data, we further included datasets for **Glucose Level** (Cho et al., 2023) and **Heart Rate** prediction (Peng et al., 1999).

The **Glucose** Cho et al. (2023) dataset measures interstitial glucose concentration of 16 subjects over the course of 10 days. We used the Dexcom G6 dataset measuring interstitial glucose concentration (mg/dL) every 5 minutes.

The meditative **Heart-Rate** dataset Peng et al. (1999) records heart-rate of 14 volunteers during a 10-minute metronomic breathing meditation session, where it is recorded as relative time since the start of the meditation session. However, some foundation models (TimesFM) require and use timestamps for forecasting so we mapped the Heart-Rate dataset to start at 2000-01-01 00:00:00 and last 10 minutes.

To evaluate models on coarser time granularities, we also use the **Patents** dataset (Marco et al., 2015) which counts the number of US patents filed per month from 1981 to 2014. We removed sparse time series, aggregated by patent field and category, and filtered out timeseries with a minimum value of less than 100.

Additional dataset statistics can be found in the partial autocorrelation plots in Figure 6. We did not evaluate the AR experiments on the Heart-Rate and Patents dataset as they did not contain enough data to accurately assess long-term forecasting calibration.

Both synthetic noise datasets were generated as a single time series with length 4367 and the train/test split at 1440. The IID noise time series was generated using $y_t = \epsilon$ and the noisy first-order linear process $y_t = \alpha y_t - 1 + (1 - \alpha)\epsilon$ where $\alpha = 0.9$ and $\epsilon$ is drawn from the standard normal distribution $\mathcal{N}(0, 1)$.

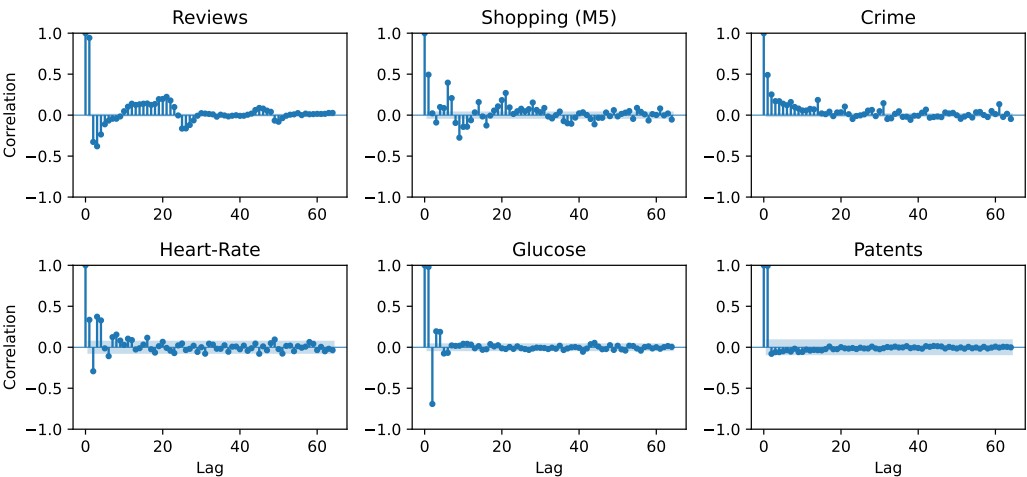

Figure 6: **Partial Autocorrelation Plots: Glucose and Patents datasets have limited number of lags with significant partial autocorrelation while having large correlation at the first lag.** The naive model is well suited for these two datasets which would explain the poor point forecasting performance (MASE). The specific time-series from each dataset used to generate the plots were selected based on having approximately median MASE scores in that dataset.

# B  ADDITIONAL RESULTS AND FIGURES

## B.1  BASE FINDINGS

We include additional figures and findings for each of the experiments below. Table 4 shows the short-term calibration and accuracy scores of each model across all of the datasets.

In Figure 11, we show that across time series in the same dataset, the model point accuracy, MASE, are not correlated with model calibration. However, across datasets as in Figure 9, we see a strong correlation between TSFM, MASE, and PCE with a correlation coefficient of 0.90. This relationship does not always hold. For example, despite the TSFMs having poor point accuracy on the Glucose dataset, worse than a naive predictor with MASE greater than 1.0, they are still well calibrated with PCE less than 0.05. We note that for multi-step predictions ($L > 1$) that the MAE of the naive predictor in the denominator of MASE is in effect using information "from the future," with the result that the overall MASE may take values greater than 1 even for useful models. In Figures 7 and 8, as the TSFMs predict further into the future, not surprisingly the forecasts become both less accurate and less calibrated. We see exceptions in Reviews where the TSFMs remain equally well calibrated up to a forecast length of 64. However, unlike the TSFMs, the baseline models have a consistently high PCE over all prediction lengths.

The TSFMs are not consistently over- nor under-confident (see Figures 3 and 12). However among the TSFMs, YingLong is more confident with tighter intervals compared to the other TSFMs, with some indication of overconfidence on Heart-Rate and Patents datasets.

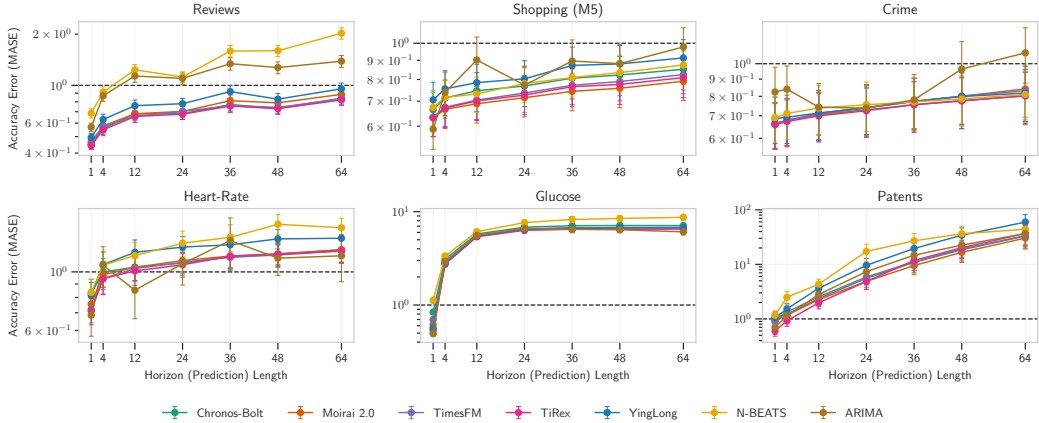

Figure 7: **TSFM point accuracy gets increasingly worse the further out the model forecasts** Mean Absolute Scaled Error (MASE) as a function of prediction length across the six datasets. The dashed horizontal line is the MASE of the naive predictor ($y_t = y_{t-1}$) as a reference.

Table 4: Calibration evaluation results. Best results are highlighted in bold, and second best results are underlined.

| | | Foundation Models | | | | | Baselines | |
|---|---|---|---|---|---|---|---|---|
| | | Chronos-Bolt | Moirai 2.0 | TimesFM | TiRex | YingLong | N-BEATS | ARIMA |
| MASE | Reviews | 0.717 | 0.748 | 0.720 | **0.710** | 0.823 | 1.416 | 1.211 |
| | Shopping (M5) | 0.785 | **0.727** | 0.753 | 0.743 | 0.835 | 0.791 | 0.854 |
| | Glucose | 6.362 | 5.986 | 6.051 | **5.841** | 6.041 | 7.326 | 5.958 |
| | Heart-Rate | 1.119 | 1.118 | 1.104 | **1.096** | 1.257 | 1.335 | 1.120 |
| | Crime | 0.741 | 0.759 | 0.759 | **0.740** | 0.761 | 0.767 | 0.829 |
| | Patents | 11.896 | **10.639** | 12.193 | 12.510 | 21.297 | 22.814 | 14.534 |
| | IID Noise | 0.706 | 0.705 | 0.707 | 0.708 | 0.709 | 0.926 | **0.704** |
| | Linear Process | 2.491 | 2.417 | 2.210 | 2.325 | 2.322 | 2.790 | **2.166** |
| PCE | Reviews | 0.015 | 0.013 | 0.012 | **0.012** | 0.014 | 0.061 | 0.176 |
| | Shopping (M5) | 0.033 | 0.045 | 0.036 | **0.031** | 0.052 | 0.108 | 0.131 |
| | Glucose | 0.036 | 0.033 | 0.050 | **0.031** | 0.042 | 0.092 | 0.141 |
| | Heart-Rate | 0.063 | 0.051 | 0.052 | **0.048** | 0.077 | 0.121 | 0.204 |
| | Crime | 0.023 | 0.035 | 0.023 | 0.021 | **0.014** | 0.103 | 0.168 |
| | Patents | 0.176 | **0.154** | 0.206 | 0.192 | 0.259 | 0.242 | 0.247 |
| | IID Noise | 0.009 | 0.008 | **0.004** | 0.012 | 0.010 | 0.072 | 0.122 |
| | Linear Process | 0.033 | 0.015 | **0.008** | 0.022 | 0.021 | 0.138 | 0.122 |
| CCE | Reviews | -0.012 | 0.007 | **0.004** | -0.013 | -0.024 | -0.040 | -0.351 |
| | Shopping (M5) | 0.002 | **0.000** | 0.005 | -0.026 | 0.051 | -0.203 | -0.259 |
| | Glucose | **-0.011** | -0.050 | -0.038 | -0.027 | 0.034 | -0.172 | -0.280 |
| | Heart-Rate | -0.061 | -0.008 | 0.001 | -0.037 | 0.105 | -0.243 | -0.411 |
| | Crime | -0.030 | -0.020 | **-0.001** | -0.041 | 0.015 | -0.207 | -0.337 |
| | Patents | **0.028** | 0.046 | 0.089 | 0.069 | 0.201 | 0.322 | 0.031 |
| | IID Noise | 0.007 | -0.008 | **0.007** | -0.012 | 0.016 | 0.111 | -0.244 |
| | Linear Process | 0.006 | -0.020 | **0.002** | -0.021 | 0.044 | 0.256 | -0.244 |
| SIW | Reviews | 0.198 | 0.204 | 0.180 | 0.195 | 0.248 | 0.573 | 1.138 |
| | Shopping (M5) | 0.240 | 0.231 | 0.241 | 0.260 | 0.233 | 0.538 | 0.644 |
| | Glucose | 0.996 | 0.972 | 1.001 | 0.911 | 0.761 | 1.727 | 1.967 |
| | Heart-Rate | 0.716 | 0.605 | 0.585 | 0.653 | 0.496 | 1.416 | 2.174 |
| | Crime | 0.278 | 0.276 | 0.261 | 0.290 | 0.246 | 0.553 | 0.852 |
| | Patents | 0.108 | 0.096 | 0.088 | 0.086 | 0.091 | 0.049 | 0.092 |
| | IID Noise | 0.991 | 1.035 | 0.999 | 1.051 | 0.977 | 1.118 | 2.026 |
| | Linear Process | 1.107 | 1.164 | 0.985 | 1.110 | 0.930 | 0.629 | 1.942 |
| WQL | Reviews | 56.732 | 59.070 | 56.929 | **56.141** | 65.569 | 98.865 | 128.830 |
| | Shopping (M5) | 59.558 | **55.129** | 57.137 | 56.454 | 63.402 | 56.402 | 63.618 |
| | Glucose | 61.234 | 57.926 | 58.157 | 56.564 | 58.398 | 64.942 | **56.257** |
| | Heart-Rate | 19.410 | 19.332 | **18.967** | 19.011 | 21.812 | 22.679 | 27.851 |
| | Crime | 39.293 | 40.129 | 40.198 | 39.426 | 40.409 | **37.809** | 47.707 |
| | Patents | 22.549 | **20.958** | 24.341 | 24.435 | 45.050 | 47.238 | 25.886 |

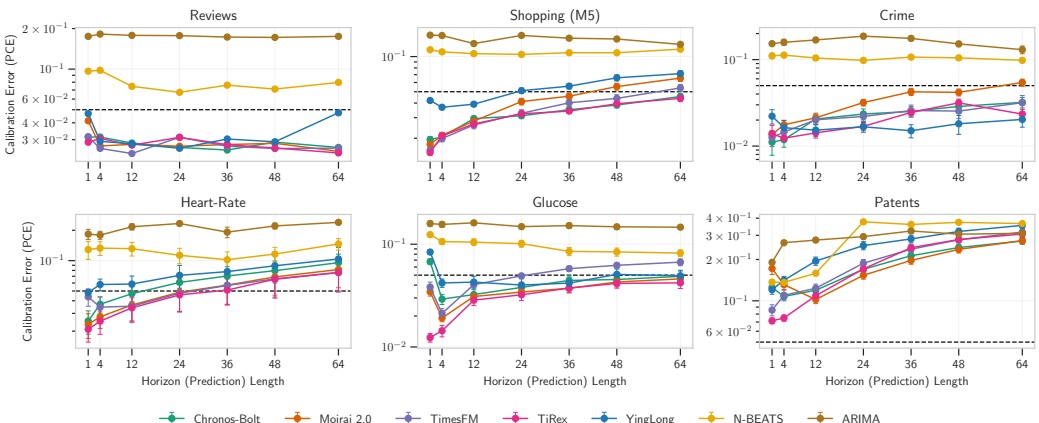

Figure 8: **TSFM calibration gets increasingly worse the further out the model forecasts** Probabilistic Calibration Error (PCE) as a function of prediction length across the six datasets. PCE values below the dashed horizontal line ($y = 0.05$) can be considered to be well-calibrated.

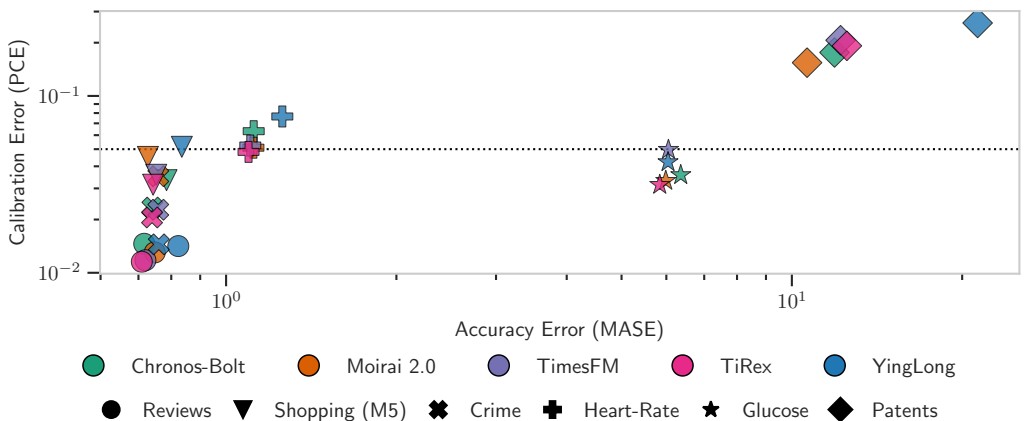

Figure 9: **Aggregated across each dataset TSFMs have a correlation between point accuracy (MASE) and calibration (PCE).** Probabilistic calibration error (PCE) versus point-forecast accuracy (MASE) averaged over all time series in each dataset.

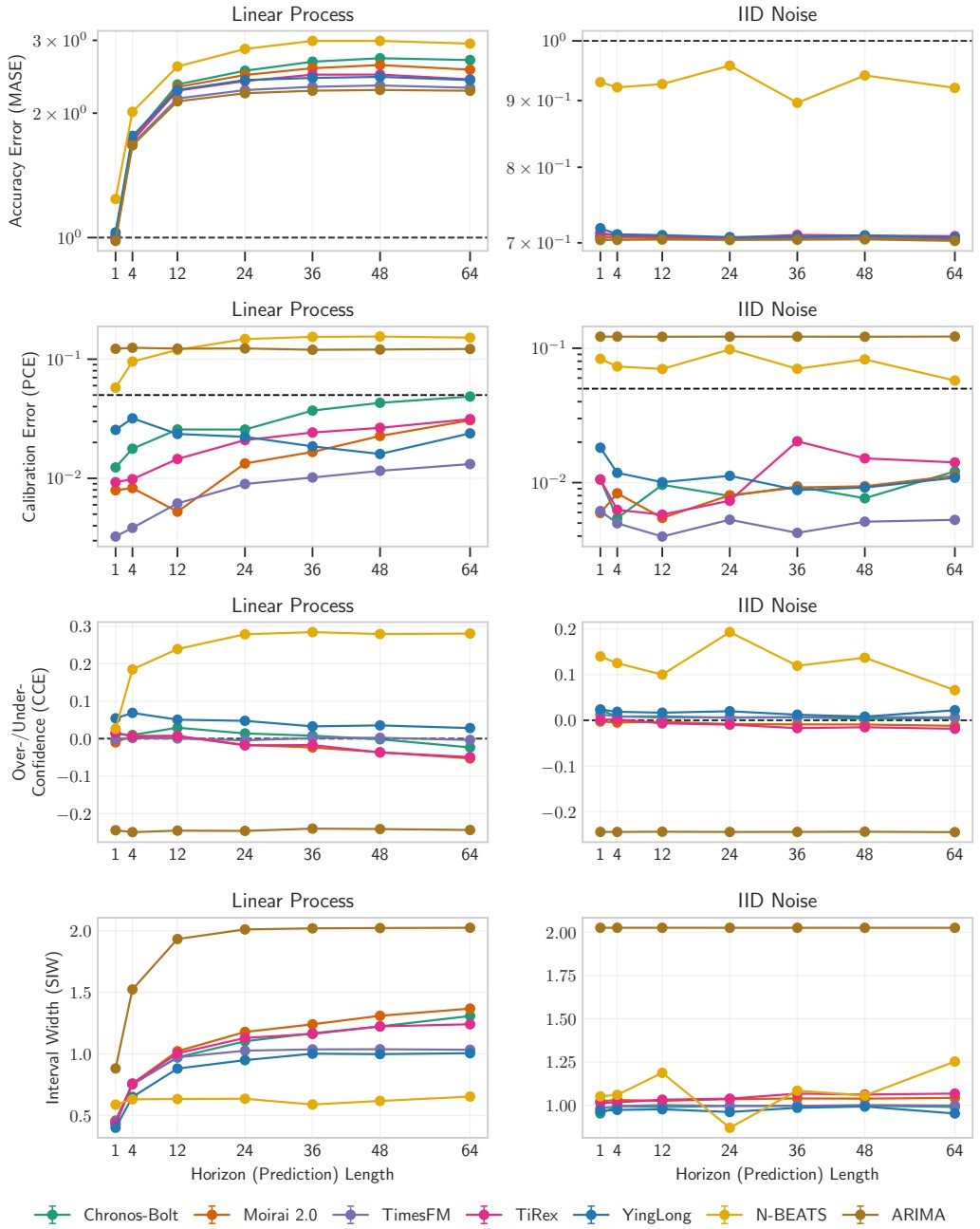

Figure 10: **TSFMs are well calibrated on synthetic datasets while ARIMA is under-confident predicting higher variance (SIW).** Plots calibration and accuracy metrics as a function of horizon length for synthetic datasets.

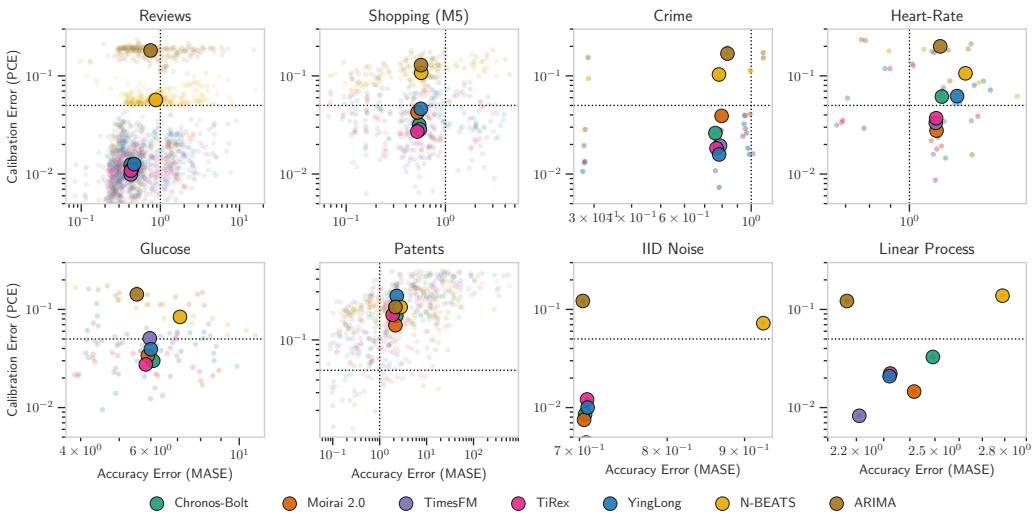

Figure 11: **TSFMs are better calibrated than the baselines. Within each dataset, there is no correlation between MASE and PCE.** Probabilistic calibration error (PCE) versus point-forecast accuracy (MASE) across the six datasets. Each dot represents model performance on an individual time series; larger centroids being the average over all time series in a dataset. PCE values below 0.05 (dotted horizontal line) are well-calibrated while MASE values less than 1.0 (dotted horizontal line) are better than the naive predictor. MASE values greater than 1.0 can still be a useful model due to the naive model using information "from the future."

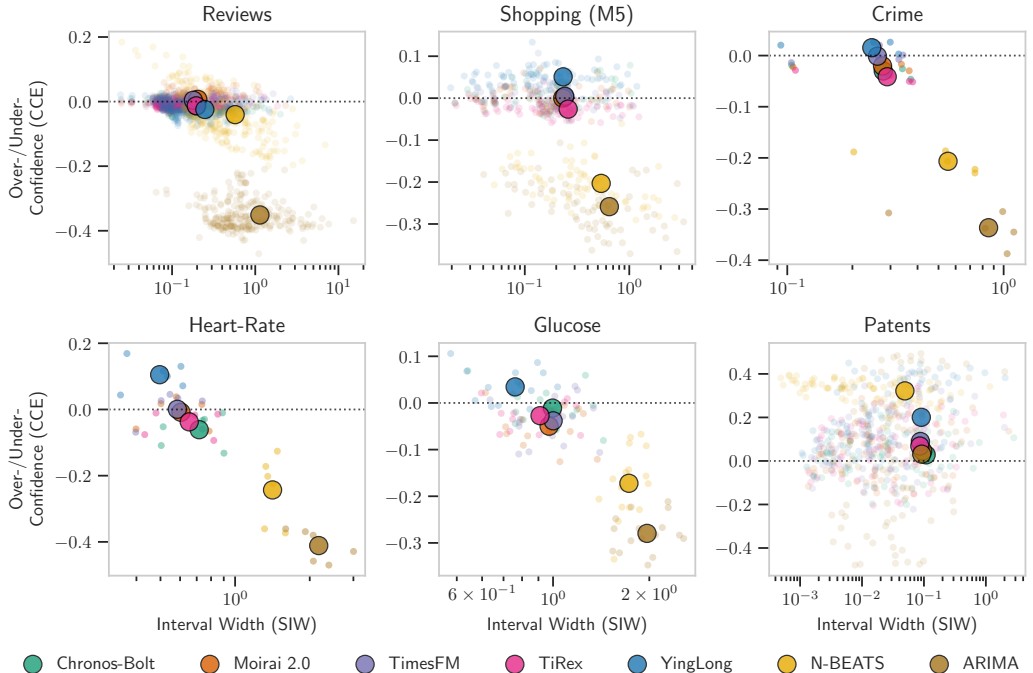

Figure 12: **TSFMs tend to be neither over/under-confident overall. Models with larger SIW, wider confidence intervals, were more under-confident.** Centered Calibration Error (CCE) versus Scaled Interval Width (SIW) across the six datasets. Each dot represents model performance on an individual time series; larger centroids being the average over all time series in a dataset.

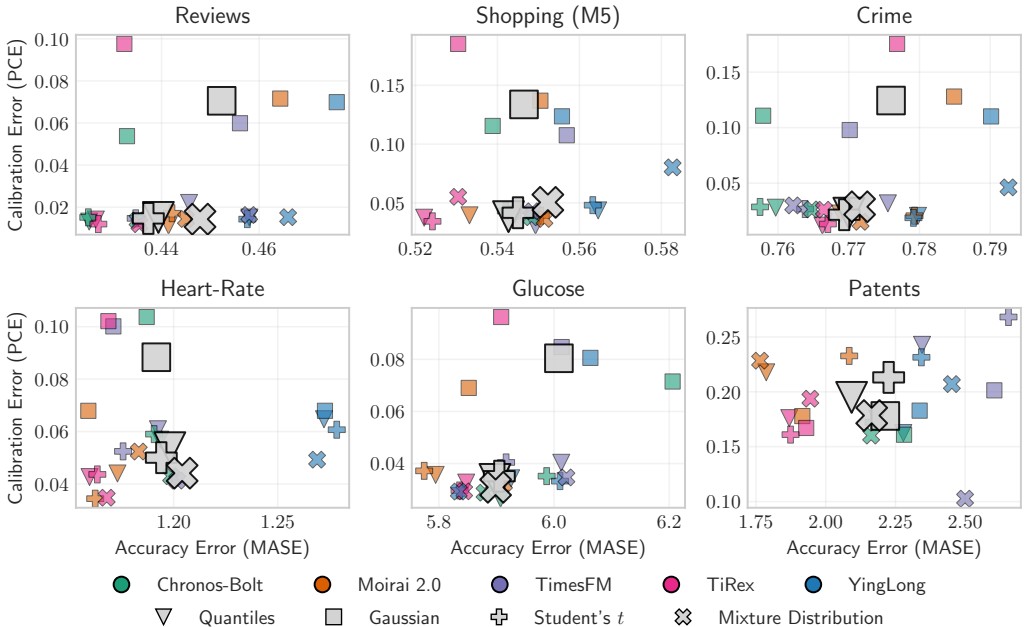

Figure 13: **All prediction heads tend to have very similar accuracy error with the Gaussian head occasionally having slightly worse error.** Calibration and accuracy properties of various prediction heads with Accuracy Error (MASE) on the $x$-axis and Probabilistic Calibration Error (PCE) on the $y$-axis. Background markers are the results for each head (shape) with each backbone model (color), while the larger gray centroids are the mean results across models.

## B.2 Comparison with Additional Metrics

As mentioned in the main paper, WQL, MSIS, and CRPS are common metrics for evaluating model calibration. Weighted Quantile Loss (WQL) is an approximation of Continuous Ranked Probability Score (CRPS) defined as the pinball (or quantile) loss $p_q$ scaled by the absolute sum of the true values:

$$p_q(y_t, \hat{y}_t^q) = \begin{cases} 2 \cdot (1 - q) \cdot (\hat{y}_t^q - y_t) & \text{if } \hat{y}_t^q \geq y_t \\ 2 \cdot q \cdot (y_t - \hat{y}_t^q) & \text{if } \hat{y}_t^q < y_t \end{cases} \tag{5}$$

$$WQL = \frac{1}{\sum_{t=T+1}^{T+L} |y_t|} \sum_{t=T+1}^{T+L} \sum_q p_q(y_t, \hat{y}_t^q) \tag{6}$$

However, as described in Chung et al. (2021), CRPS and WQL, measure a combination of probabilistic calibration and sharpness Gneiting et al. (2007). This combination leads to an imbalance often skewing to prioritize predictive sharpness Chung et al. (2021).

Mean Scaled Interval Score (MSIS) is a scaled version of Mean Interval Score (MIS) which is the mean difference in upper and lower bound prediction penalized with the error when the true value lies outside the bounds:

$$\begin{aligned} MSIS = \frac{1}{MAE_n} \frac{1}{L} \sum_{t=T+1}^{T+L} & (\mathcal{U}_t^s - \mathcal{L}_t^s) \\ & + \frac{2}{1 - s} (\mathcal{L}_t^s - y_t) \mathbf{1}[y_t < \mathcal{L}_t^s] \\ & + \frac{2}{1 - s} (y_t - \mathcal{U}_t^s) \mathbf{1}[y_t > \mathcal{U}_t^s] \end{aligned} \tag{7}$$

MSIS has the same limitations being a measure of interval size with a penalty term for observed values outside the interval Gneiting et al. (2007); Hyndman & Athanasopoulos (2018); Gneiting & Raftery (2007). We find that these metrics were highly correlated to MASE in Figure 15. Therefore, when using these metrics to evaluate calibration, their values result in a measure of sharpness and accuracy that diverge from a measurement of calibration (see Figure 14). For example, if we evaluate calibration using WQL or MSIS, we would incorrectly conclude that ARIMA is equally or better calibrated than the best foundation models on the Glucose dataset as in Figure 1 and 14.

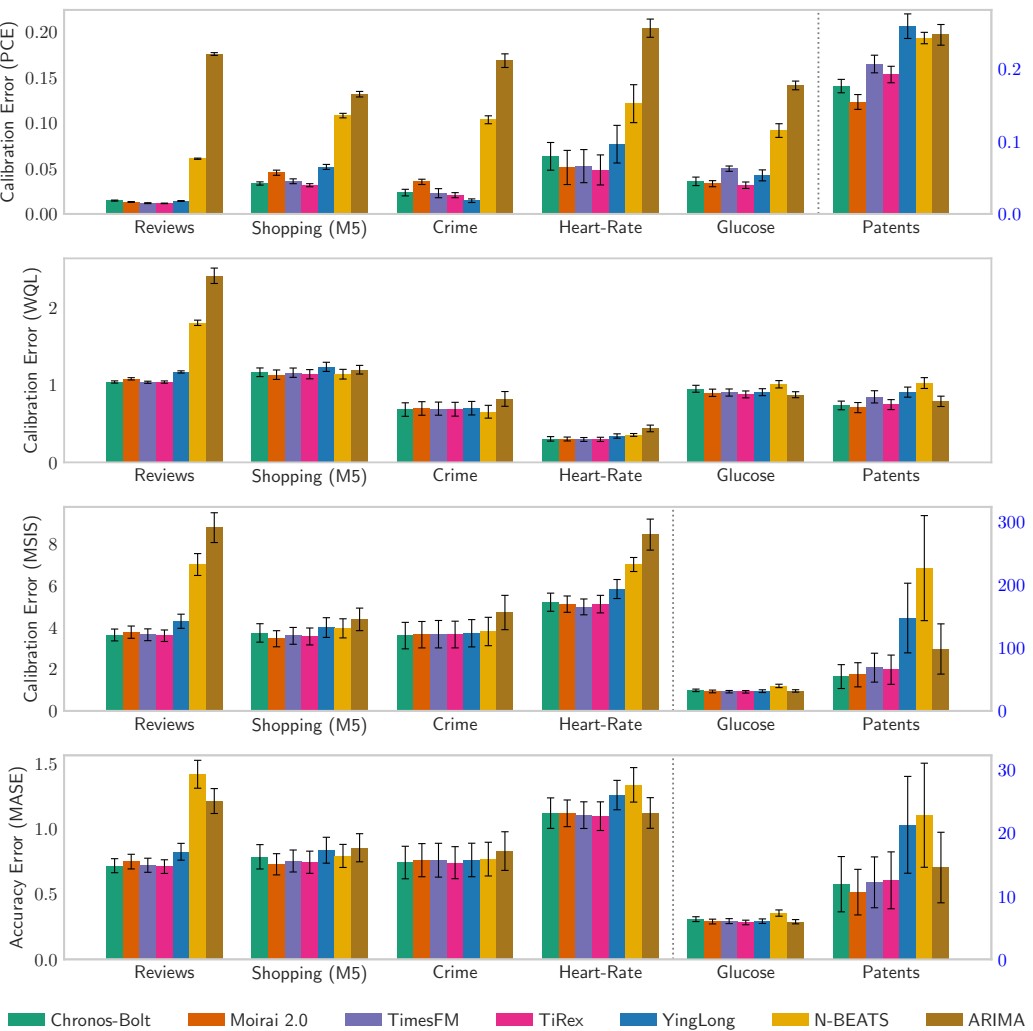

Figure 14: **WQL and MSIS incorrectly imply that N-BEATS and ARIMA are well calibrated on some datasets, while PCE indicates they are always poorly calibrated.** Top: Probabilistic Calibration Error (PCE) across datasets and models using the default quantile prediction head, Patents PCE uses the right $y$-axis scale. Upper Middle: Weighted Quantile Loss (WQL) measuring a combination of sharpness and calibration. Lower Middle: Mean Scaled Interval Score (MSIS) where Glucose and Patents use right $y$-axis scale. Bottom: Mean Absolute Scaled Error (MASE) measuring point accuracy of median prediction, Glucose and Patents use the right $y$-axis scale.

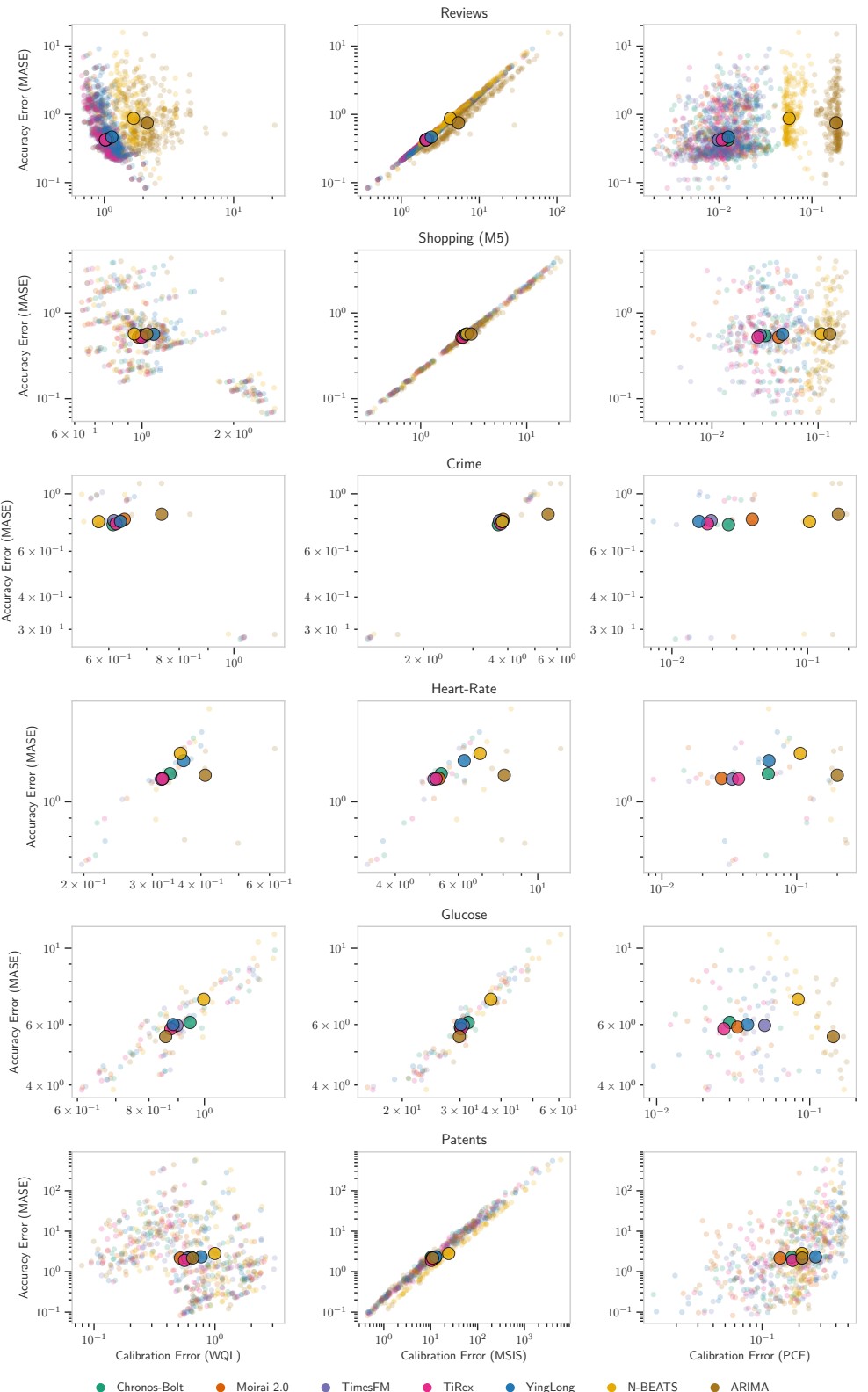

Figure 15: **WQL and MSIS are often highly correlated with MASE across time series within a dataset** WQL (left), MSIS (middle), and PCE (right) compared to MASE where each dot is the performance of an individual time series.

### B.3 TAIL FORECASTING

In downstream tasks such as Anomaly Detection, calibration at the tail ends of probabilistic predictions is more important than at the body of a predictive distribution. We evaluate the tailed calibration with a modified PCE and CCE that only considers the 0.1 and 0.9 quantile predictions rather than averaging over all the quantiles or confidence intervals. The Tailed PCE (TPCE) and Tailed CCE (TCCE) values are not directly comparable to their aggregate values as their scales could be slightly offset. Models that are in general overconfident with positive CCEs tend to be have an exaggerated overconfidence and miscalibration at the tails. The opposite is true with under-confident models, having improved calibration at the tails while their under-confidence CCE is capped and reduced by being at the tail. In Figure 16, we find that YingLong which is overconfident with overly tight confidence intervals, are relatively poorer at the tails of the distribution compared to the other models, while ARIMA sees a relative improvement. For the TSFM models that are neither over nor under-confident, their calibration looks to be improved with very minor calibration error. Promising future work should evaluate tailed calibration with quantiles further on the tail than 0.1 or 0.9 such as 0.001 or 0.999.

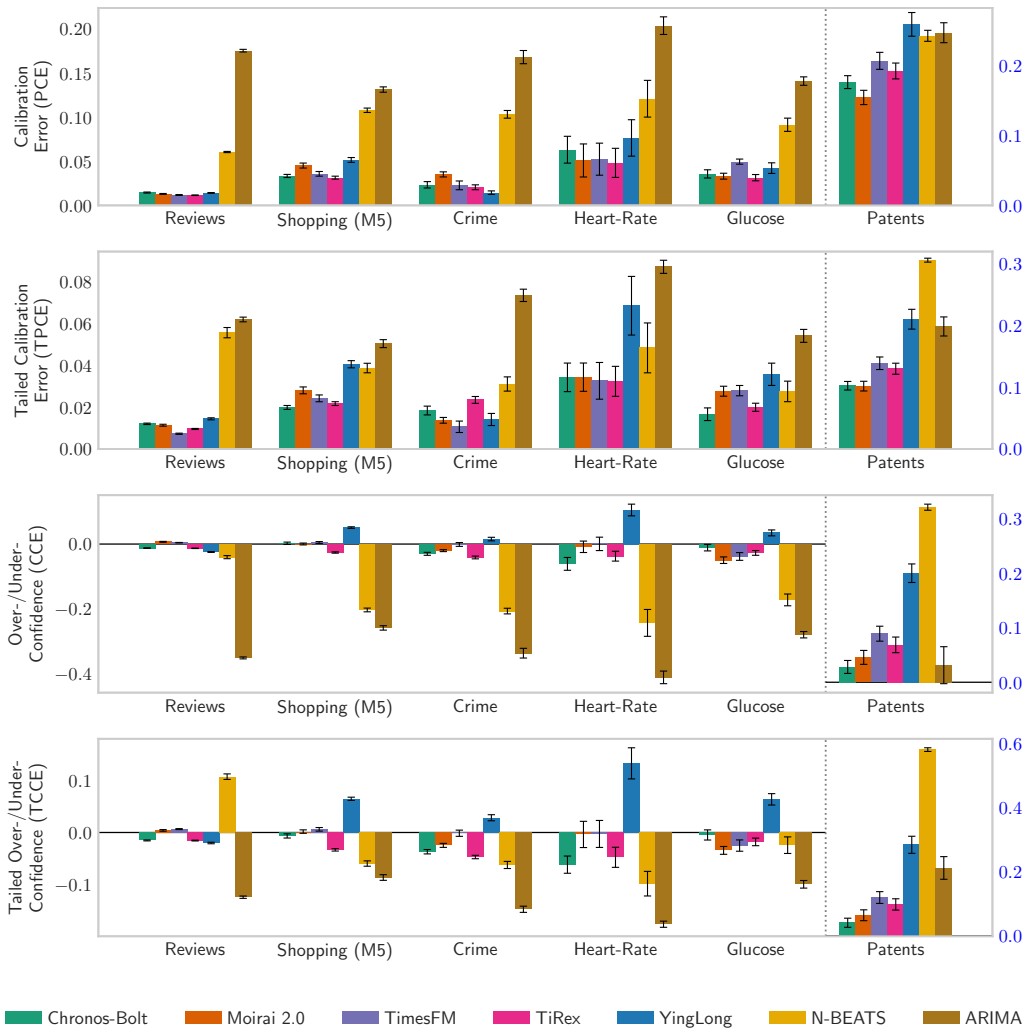

Figure 16: **Tailed Calibration Error (TPCE) is generally smaller than Calibration Error (PCE) while Tailed Over-/Under-Confidence (TCCE) grows for overconfident models and reduces for under-confident models.** Comparison of model calibration on the entire probabilistic distribution versus calibration of the tail-ends of the distribution (0.1 and 0.9).

## B.4 ADDITIONAL FINDINGS FOR AR FORECASTING

In this section, we elaborate on two key issues that arise when choosing horizon length in autoregressive time series forecasting models (TSFMs). Compounding errors degrade accuracy, and blockwise independence introduces bias in the input for future predictions.

1. Autoregressive errors compound. When forecasting multiple time points autoregressively, each predicted value is fed back as input for predicting subsequent points. Even small errors in early predictions can accumulate over the horizon, leading to progressively degraded forecasts. This is exacerbated when using smaller horizons as the error can accumulate faster.

2. Independent block modeling introduces bias. TSFMs model a block of $H$ future time points as independent given past context:

$$p(x_i, \ldots, x_{i+H-1}|x_{<i}) \approx \prod_{j=i}^{i+H-1} p(x_j|x_{<i}).$$

While convenient for training, this ignores any intra-block dependencies in longer horizons. During autoregressive rollouts, the next block of predictions is conditioned on samples from the previous block. Because the sampled block does not preserve the true joint dependencies, the model sees a biased context, resulting in a shift in the marginal distribution of later time points. This bias is larger for longer blocks $H$ (since then there are fewer biased inputs) and vanishes as $H \rightarrow h$ (where each time point is predicted independently in an autoregressive manner).

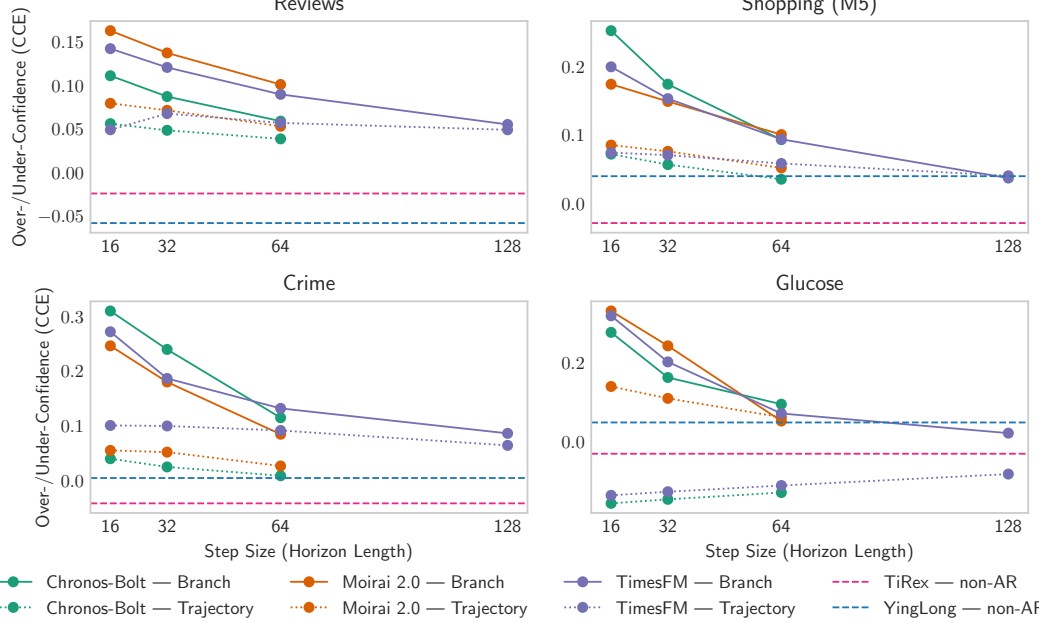

Figure 17: **TSFMs are consistently overconfident on long-term forecasting. The magnitude of the over confidence decreases with longer horizon lengths and when using the trajectory AR method.** The figure compares Centered Calibration Error (CCE) for long-term forecasting using autoregression on the $y$-axis with AR prediction horizon on the $x$-axis. The color of the line depicts the model used and the line style indicates the AR method. The pink and blue dashed horizontal lines are the CCE for TiRex and Yinglong without using AR.

## B.5 ADDITIONAL MIXTURE DISTRIBUTION HEADS

In addition to the four prediction heads included in the main paper, Quantiles, Gaussian, Student's $t$, and Moirai 1.0 mixture distribution, we include experiments with two additional mixture heads:

mixture of Gaussian distributions and mixture of Student's $t$ distributions. Each mixture head contains five of their respective distributions.

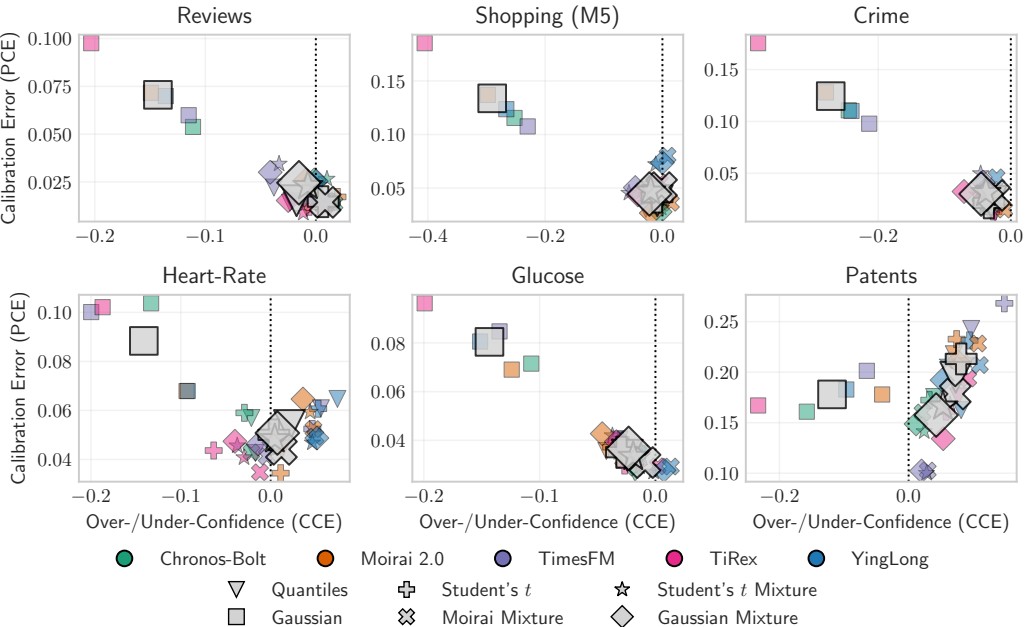

Figure 18: **Both mixtures of Gaussian and Student's $t$ distribution heads maintain a similarly low calibration error as the other non-Gaussian heads from the main paper.** Calibration properties of various prediction heads with Centered Calibration Error (CCE) on the $x$-axis and Probabilistic Calibration Error (PCE) on the $y$-axis. Large positive CCE values indicate overconfident predictions, while large negative CCE values show under-confidence. Background markers are the results for each head (shape) with each backbone model (color), while the larger gray centroids are the mean results across models.

### B.6 ADDITIONAL RESULTS ON GIFT-EVAL

To further supplement our experiments, we evaluated the TSFMs on a subset of the GIFT-EVAL dataset (Aksu et al., 2024). Due to the GIFT-EVAL dataset containing heterogeneous data with inconsistent time series lengths and unbalanced sub-datasets, we filtered out time series with lengths less than 768 points (512 context and 256 prediction length) and sampled a max of 1024 time series from each sub-dataset. In Figure 19, we show that the calibration results on this subset of the GIFT-EVAL dataset align with the results from the datasets included in the main paper.

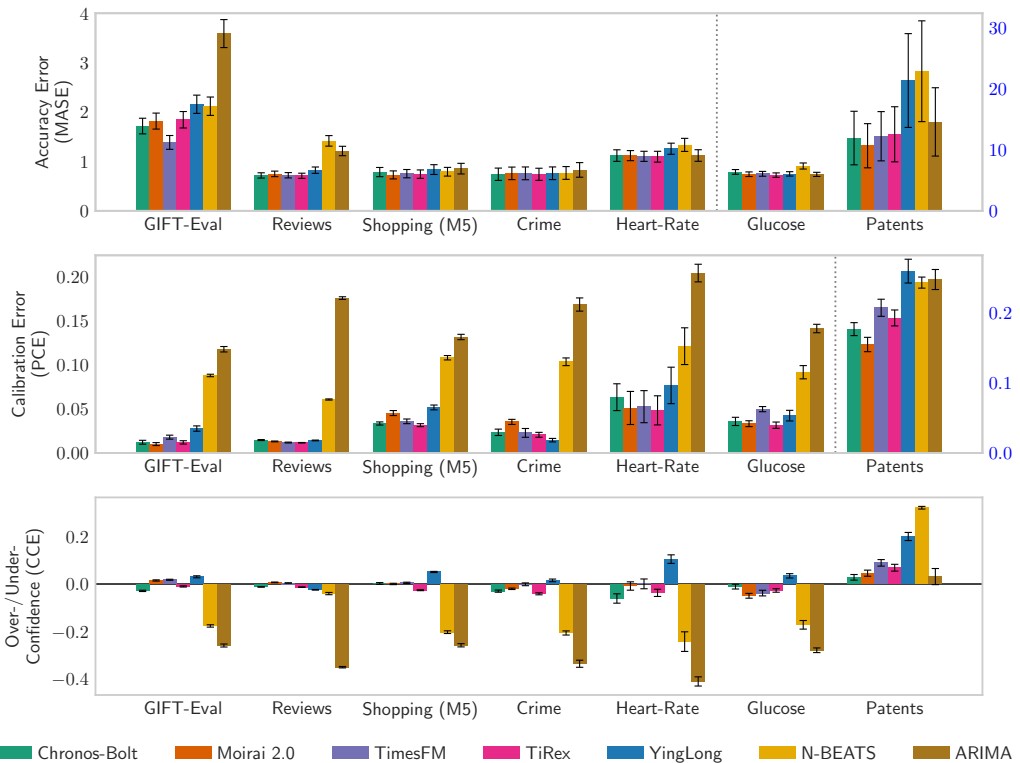

Figure 19: **The TSFMs have a significantly reduced calibration error compared to the baseline methods on the GIFT-EVAL dataset similar to the other datasets.** Top: Mean Absolute Scaled Error (MASE) measuring point accuracy error of the median prediction (lower is better), Glucose and Patents use their own $y$-axis scale (on the right). Middle: Probabilistic Calibration Error (PCE) across datasets and models using the default quantile projection block. (Patents PCE uses its own $y$-axis scale (on the right)). Lower PCE values are better. Bottom: Centered Calibration Error (CCE) evaluating systematic overconfidence (positive) or under-confidence (negative).

