# OpenReview forum: "Beyond Accuracy: Are Time Series Foundation Models Well-Calibrated?"
_ICLR.cc/2026/Conference — ICLR 2026 Poster_

### Official Review · Reviewer_cuh6 · 2025-10-31

**Soundness:** 3
**Presentation:** 3
**Contribution:** 3
**Rating:** 6
**Confidence:** 3

**Summary:**

This paper investigates the calibration of time series foundation models. The study analyzes five foundation models and two non-pretrained baseline models, evaluating their calibration performance, the impact of different prediction heads, and calibration under long-term autoregressive forecasting. The findings reveal that time series foundation models consistently demonstrate superior calibration compared to baseline models, exhibiting neither systematic over-confidence nor under-confidence.

**Strengths:**

- The paper is well written and easy to follow.
- The problem is interesting: calibration is an important problem, yet not commonly discussed by the community.
- The evaluation metrics set in the paper are comprehensive, and a series of very interesting conclusions have been drawn.

**Weaknesses:**

- This is a highly experimental paper with substantial experimental content. But it lacks a deep analysis. For example, in Section 4.4, the author's experimental results indicate that "the predictions with a shorter forecast horizon $p$ have poorer calibration", yet no thorough analysis is provided. It would be interesting to formally know how the "horizon length" affects the prediction performance and why.

**Questions:**

- In Equation 3, it appears that a curly brace is missing to enclose all the content after the $\sum_{s \in S}$.
- In Section 4.3, the authors concluded that the calibration performance of the Gaussian distribution is inferior to that of other distributions. Could you provide a more in-depth explanation for this phenomenon? Generally, errors are widely assumed to follow a Gaussian distribution. But why is the calibration performance of the Gaussian distribution the poorest?

---

> ### Author Response · Authors · 2025-11-20
> **Rebuttal by Authors**
>
> We would like to thank the reviewer for their constructive comments and questions.
> > This is a highly experimental paper with substantial experimental content. But it lacks a deep analysis.
>
> We agree that our paper is focused on experimental results rather than deep analysis: however, we would like to point out to the reviewers and AC that there is a long history in machine learning of important and influential experimental papers [1,2] that brought the attention of the community to an important ML problem from an experimental perspective, providing a starting point for subsequent algorithmic and theoretical work on the topic. As mentioned in our responses to other reviewers, we will add more discussions on potential factors that could affect model calibration and future directions to the paper.
>
> [1] Guo, Chuan, et al. "On calibration of modern neural networks." International Conference on Machine Learning. PMLR, 2017.
>
> [2] Sokolova, Marina, Nathalie Japkowicz, and Stan Szpakowicz. "Beyond accuracy, F-score and ROC: a family of discriminant measures for performance evaluation." Australasian joint conference on artificial intelligence. Berlin, Heidelberg: Springer Berlin Heidelberg, 2006.
>
> > It would be interesting to formally know how the "horizon length" affects the prediction performance and why.
>
> Longer horizon lengths $p$ result in better calibration in autoregressive forecasts.
>
> There are two conceptual problems with short horizons: (1) Autoregressive errors can compound: feeding the generated sequences back into the model, deteriorating the quality. (2) TSFMs treat time points within the horizon approximation independently: when sampling a block of length $L$, intra-block dependencies are ignored, so the next block is conditioned on a biased context. This introduces a bias in the marginal distribution of later time points, which vanishes for larger block sizes. We have updated the manuscript with a deeper explanation of this finding in Appendix B.4.
>
> > Why is the calibration performance of the Gaussian distribution the poorest?
>
> While we do not have a definitive answer to this question, the Gaussian distribution is strictly less expressive than all other prediction heads. We speculate that this inability to properly model the uncertainty limited the probabilistic forecasting performance. A more in-depth analysis is an ideal direction for future work (which we will mention in the Discussion), but we believe this is beyond the scope of the current paper.

---

### Official Review · Reviewer_dAcF · 2025-10-31

**Soundness:** 2
**Presentation:** 3
**Contribution:** 2
**Rating:** 4
**Confidence:** 4

**Summary:**

The paper targets an underexplored but crucial aspect of time-series foundation models (TS FMs): calibration. It conducts a systematic evaluation of five recent TS FMs against two strong baselines, examining (i) overall calibration and confidence (over/under), (ii) the effect of different prediction heads, and (iii) calibration behavior under long-horizon autoregressive forecasting.

**Strengths:**

1. The authors investigate the calibration properties in time series foundation models. The topic is of interest in the community.
2. The experiments are comprehensive and confirm that time series foudnation models are well-calibrated.

**Weaknesses:**

1. Although it is an interesting topic and well motivated, but the manualscript does not provide technicial contributions. The main finding is that time series foundation models are well-calibrated compared with traditional and machine learning / deep learning models. It would be better that if we have some contributions on how to further improve the calibration ability.

**Questions:**

1. For different prediction heads, it looks like they are similar in terms of calibration error except Gaussian prediction heads. But I wonder how the forecasting performance changes with different prediction heads. Because I feel if the calibration errors are similar, then we should choose the one with better forecasting performance when training a time series foundation model.

---

> ### Author Response · Authors · 2025-11-20
> **Rebuttal by Authors**
>
> We thank the reviewer for their feedback.
>
> > It would be better if we have some contributions on how to further improve the calibration ability.
>
> The community currently lacks a systematic understanding of how modern time-series foundation models behave with respect to probabilistic calibration across diverse datasets and tasks. This is what our paper provides, and, most importantly, we find that TSFMs are surprisingly well-calibrated. We also identify important practical consequences: (a) Gaussian prediction heads are badly calibrated, (b) Long-range autoregressive predictions benefit from long-range model predictions, and (c) TSFMs are not systematically over- or under-confident.
>
> > I wonder how the forecasting performance changes with different prediction heads.
>
> Interestingly, the forecasting performance is less affected by the prediction head than the calibration. We have generated an updated accuracy (MASE) figure for models with different prediction heads, which can be found [here](https://imgur.com/a/ERUYbHu) and will be added as a figure in the Appendix.
>
> As shown in the linked figure, there is not one clear prediction head with the best forecasting accuracy. While the Gaussian head is consistently worse calibrated than the other prediction heads, it has similar accuracy on the Shopping (M5), Heart-Rate, and Patents datasets with slightly worse accuracy on Reviews, Crime and Glucose datasets.

---

### Official Review · Reviewer_NKRH · 2025-11-01

**Soundness:** 3
**Presentation:** 3
**Contribution:** 3
**Rating:** 8
**Confidence:** 3

**Summary:**

This paper presents the first systematic empirical study examining the calibration properties of time series foundation models (TSFMs). Using appropriate metrics and datasets, the authors first evaluate the forecasting calibration levels of TSFMs, finding that they outperform traditional statistical models and exhibit better calibration. Furthermore, the paper investigates the effect of prediction head design on calibration, revealing that Gaussian heads tend to be under-confident in most cases and are outperformed by alternative head types. The study also explores the impact of forecasting horizon length on calibration, showing that for long-term autoregressive forecasting, increasing the horizon length and using the trajectory AR method leads to better-calibrated forecasts.

**Strengths:**

1. Appropriate experiment setting: The work identifies the shortcomings of commonly used metrics (CRPS, WQL, MSIS) that conflate calibration and sharpness. By prioritizing PCE and CCE, the paper provides a clearer measure of true probabilistic calibration.

2. Novel insights: The study offers meaningful analysis and insights in each experiment. In particular, it systematically investigates the calibration properties of TSFMs, and explores how factors such as prediction head design and forecasting horizon influence calibration—topics that have not been examined in prior work

3. Clear organization: The paper is well-structured, with logically arranged sections and a coherent flow, making it easy for readers to follow the results and key takeaways.

**Weaknesses:**

1. Section 4.3 and 4.4 lack clear motivation for investigating the influential factors affecting calibration. Why did the authors choose to focus on the impact of prediction head design and forecasting horizon, rather than other factors such as context length or data characteristics ?

2. The investigation of the calibration properties of TSFMs remains at the validation stage and does not further explore the underlying reasons why TSFMs tend to be well-calibrated, whereas traditional statistical models are often overconfident. A deeper analysis regarding the mechanisms behind this difference would make the work more insightful.

3. While the finding of general insensitivity to the prediction head form is valuable, the scope of prediction head selection is limited. Recently, more novel designs like flow-matching prediction head [1] could also offer new insights into calibration behavior.

[1] Liu Y, Qin G, Shi Z, et al. Sundial: A family of highly capable time series foundation models. arXiv preprint arXiv:2502.00816, 2025.

**Questions:**

1. Do inherent data characteristics such as randomness and autocorrelation influence the calibration evaluation of TSFMs?

2. The paper finds that TSFMs are not systematically over- or under-confident, which contrasts with observations in image and text domains. What is the possible reason for this difference?

3. A low calibration error does not correspond to a low forecasting error, as seen in the Heart Beat and Patents datasets. What causes this discrepancy between the metrics?

4. What could be the underlying reason that all autoregressive TSFMs are consistently overconfident in long-term forecasting? Whether are there measures could be taken during training to mitigate this bias?

---

> ### Author Response · Authors · 2025-11-20
> **Rebuttal by Authors**
>
> We thank the reviewer for their encouraging feedback and detailed comments.
>
> > Why did the authors choose to focus on the impact of prediction head design and forecasting horizon, rather than other factors such as context length or data characteristics?
>
> Thank you for the question! Calibration evaluation with respect to the prediction head is especially relevant as the prediction head defines how the model computes the probabilistic forecasts. Additionally, it is important to identify whether the choice of prediction head may bottleneck probabilistic predictions or if the model’s calibration is mainly dependent on the quality of the backbone.
>
> While the forecast horizon is an important aspect of long-term forecasting, we agree that there are other components that can affect calibration such as context length and will add these additional factors as potential for interesting future directions in our Discussion.
>
> > The scope of prediction head selection is limited.
>
> As with our response to Reviewer c1Kq, we agree our evaluation would benefit from additional prediction heads. We will include further prediction head evaluation results using Gaussian and Student’s $t$ mixture models to the camera-ready version of the paper. We chose to focus on additional datasets first, as suggested by reviewer c1Kq, where the new results are consistent with our original findings.
>
> > Do inherent data characteristics such as randomness and autocorrelation influence the calibration evaluation of TSFMs?
>
> We found no clear relationship between data characteristics (i.e. noise and autocorrelation) and TSFM calibration. In Section 4.1 and Figure 10 in the Appendix, we evaluated TSFMs on an IID Noise dataset and found that all TSFMs were well calibrated despite the high-levels of unpredictable noise. Additionally, we include partial autocorrelation plots in Figure 6 which indicate that each dataset we used had a unique partial autocorrelation with no clear patterns with calibration.
>
> > What could be the underlying reason that all autoregressive TSFMs are consistently overconfident in long-term forecasting?
>
> A potential reason for AR methods being over confident is due to the resetting of confidence based latent information on subsequent AR forecasts. Because it assumes the context is the ground truth it cannot reliably recover the potential error-debt that has been accumulated from prior forecasts.
>
> Solutions to this have been proposed such as not using AR forecasts as seen by encoder based models like YingLong or ones that allow for extended forecast horizons by shifting the context and using null-tokens to pad the context as in TiRex. TimesFM 2.5 [1] (a more recent update in September 2025 to the evaluated TimesFM 2.0) claims to improve calibration while continuing to use AR forecasts by generating an extra long *error forecast* at the first prediction step that can then be used along with the subsequent AR mean forecasts to build a calibrated probability forecast that does not reset the confidence information. These would be interesting future directions for in-depth investigation.
>
>
> [1] Das, et al. "TimesFM 2.5 Update." GitHub: https://github.com/google-research/timesfm?tab=readme-ov-file#update---sept-15-2025.

---

### Official Review · Reviewer_c1Kq · 2025-11-04

**Soundness:** 3
**Presentation:** 2
**Contribution:** 2
**Rating:** 4
**Confidence:** 4

**Summary:**

The paper performs an empirical study on whether time series foundation models are calibrated. They select 5 TSFMs, Chronos-bolt, Moirai 2.0, YingLong, TimesFM and Tirex, as well as N-BEATS and ARIMA as baselines. Then, paper evaluates these models on 6 datasets, focusing on calibration metrics, specifically using Probabilistic Calibration Error, Scaled Interval Width (to evaluate sharpness), and Centered Calibration Error. Experiments show that TSFMs have better calibration than baselines, and a few other insights are presented.

**Strengths:**

The paper explores an understudied aspect of time series foundation models. It investigates a wide range of recent models across 6 different datasets. Ideas are well presented and the relevant experiments have been performed.

**Weaknesses:**

* Given that the field has been mostly motivated by accuracy metrics, it would be good to give more motivation in the introduction about why researchers should be interested in calibration.
* It would be good to perform a larger scale investigation, for example computing these metrics for GIFT-eval, to get a more comprehensive understanding.
* It would be good to add in mixture of gaussians and mixture of student-ts for Fig 4, given that Toto has been mentioned several times in the paper, and Toto uses mixture of student-t.
* Section 4.4 is quite confusing, I am unclear why forecasts are more calibrated as horizon length increases. Fig 8 in the appendix also seems to contain contradictory findings. Also, why are tirex and yinglong straight lines?

**Questions:**

See weaknesses section

---

> ### Author Response · Authors · 2025-11-20
> **Rebuttal by Authors**
>
> We thank the reviewer for the helpful comments and suggestions. Please find our responses below.
>
> > It would be good to give more motivation in the introduction about why researchers should be interested in calibration.
>
> Calibration is crucial when applying models to critical domains such as medicine [1, 2], climate [3], ecology [4], and finance [5]. Without uncertainty intervals, alternative predictions are ignored, leading to overconfident decisions that can amplify risk and undermine the reliability of downstream systems. For example, consider a medical setting where a patient’s time series exhibits a significant but non-majority probability of evolving into a critical state. A point predictor may confidently output a benign trajectory, obscuring the plausible adverse outcome. Well-calibrated uncertainty intervals, by contrast, would surface this risk and enable clinicians to intervene proactively.
>
> [1] Dusenberry, Michael W., et al. "Analyzing the role of model uncertainty for electronic health records." Proceedings of the ACM Conference on Health, Inference, and Learning. 2020.
>
> [2] Harutyunyan, Hrayr, et al. "Multitask learning and benchmarking with clinical time series data." Scientific Data 6.1 (2019): 96.
>
> [3] Vitart, Frédéric, and Andrew W. Robertson. "The sub-seasonal to seasonal prediction project (S2S) and the prediction of extreme events." NPJ Climate and Atmospheric Science 1.1 (2018): 3.
>
> [4] Denny, Mark W., et al. "On the prediction of extreme ecological events." Ecological Monographs 79.3 (2009): 397-421.
>
> [5] Drechsler, Itamar. "Uncertainty, time‐varying fear, and asset prices." The Journal of Finance 68.5 (2013): 1843-1889.
>
> > The paper would benefit from a larger scale evaluation e.g. GIFT-Eval...
>
> Great suggestion! We find that TSFMs are also well-calibrated on the GIFT-Eval dataset:
>
> Updated bar plots with the new experiments can be found [here](https://imgur.com/a/8xW0yo1), and we will update the rebuttal version of the paper with the full results. In summary, we found consistent results between GIFT-Eval and our original selections of datasets.
>
> We note that this evaluation is currently performed on a subset of the GIFT-Eval dataset, and that Tirex results are still pending due to changes in their codebase we have to integrate.
>
> > It would be good to add in mixture of gaussians and mixture of student-ts
>
> We agree that adding these two heads, in addition to the current four heads (Quantiles, Gaussian, Student’s $t$, Mixture Distribution), would lead to a more comprehensive evaluation. We conjecture the results would be consistent, as our selected prediction heads are expressive. We will include full results with the two additional heads in the camera-ready version of the paper.
>
> > Section 4.4 is confusing and am unclear why forecasts are more calibrated as horizon length increases.
>
> Let’s disentangle this: The *forecast (or prediction) length* $H$ is the total number of time steps we want to predict. The model’s *forecast horizon* $p$ is the number of time steps the model predicts in a single forward pass. Therefore, when the desired *forecast length* is greater than the model’s *forecast horizon*, a model must rely on additional techniques (i.e. autoregressive forecasting) to extend the forecast to the desired length.
>
> So for Section 4.4, we use a set *prediction length* of $H=256$. As the *horizon length* $p$ increases, the model can maintain its probabilistic information for longer without it being reset in subsequent autoregressive (AR) steps. A shorter *horizon length* results in the model having its probabilistic information reset more often, as it would require more AR steps to forecast the same *prediction length*. Therefore, we would expect models with longer *horizon lengths* to have more calibrated forecasts. We will also add these discussions to the paper.
>
> > Fig 8 in the appendix also seems to contain contradictory findings.
>
> They do not contradict each other. Section 4.4 finds that if we want to predict $H$ time points into the future, calibration is better if we choose $p$ (number of time points predicted in a single pass) that is as large as possible. Figure 8 shows that models do a better job at predicting close time points (small $p$) than time points further into the future (big $p$).
>
> > Why are Tirex and Yinglong straight lines in Figure 5?
>
> Tirex and Yinglong are included as a constant baseline. The architecture of Tirex and Yinglong enables them to produce variable long-term forecast horizons in a single forward pass. Therefore, we did not evaluate their calibration using autoregressive (AR) methods. In Figure 5, we include the calibration error of these models as a baseline to compare against the AR models. Thank you again for bringing up the confusion, we will update the paper to clarify this point.

---

### Author Response · Authors · 2025-12-04
**Rebuttal Summary**

We would like to thank the reviewers for their insightful feedback and the Area Chair for their time. All reviewers (c1Kq, NKRH, dAcF, cuh6) agreed that our paper “**targets an underexplored and crucial field**” with **novel and meaningful insights**. Additionally, reviewers (NKRH, dAcF, cuh6) found that our series of **experiments and evaluation metrics were comprehensive**. Reviewers (c1Kq, NKRH, cuh6) expressed that the presentation of our **paper was clear and well-structured** making it easy for readers to follow.

We appreciate the detailed feedback from the reviewers and provided responses to individual reviewers' concerns and questions in our rebuttal. Below, we summarize key concerns from the reviewers and our responses:

- **Additional Datasets**: Reviewer c1Kq suggested a large-scale investigation on GIFT-Eval, where we observed consistent results across all our evaluated models; this strengthens our findings that time series foundation models (TSFMs) are generally well-calibrated.
- **Terminology Clarification**: In response to Reviewer c1Kq, we clarified terminology and our experimental results (Section 4.4) by clearly distinguishing between (i) the forecast (or prediction) length of users’ interests and (ii) the model’s forecast horizon that affects their calibration behaviors.
- **Empirical Contributions**: A common concern across reviewers (NKRH, dAcF, cuh6) is that our paper is highly empirical. While we agree that our paper focuses on experimental findings rather than a deeper analysis or technical contribution, we would like to point out that our paper serves as an important experimental perspective that is essential for further research towards TSFM calibration. We provide significant contributions finding that TSFMs are well-calibrated and not systematically over- or under-confident, Gaussian prediction heads are poorly calibrated, and long-range autoregressive forecasts benefit from long-range model predictions.

---

### Meta-Review · Area_Chair_NszK · 2026-01-07

**Summary:**

This paper studies the calibration properties of time series foundation models (TSFMs), which is an important and timely topic, especially given the increasing deployment of probabilistic forecasting models in real-world decision-making scenarios. All reviewers agree that the problem itself is meaningful, and the paper is generally well written.

However, for evaluation-centric papers, the community generally expects a very strong experimental scope, both in terms of datasets and target models. Unfortunately, the datasets used in this study are relatively limited and small-scale. Only five recent time series foundation models and two other baselines are covered as target models. These leads to that the empirical evidence is not sufficiently strong or comprehensive to support broad claims about the calibration properties.

Besides, multiple reviewers converged on “useful experiments, but lacks deeper analysis / technical contribution”. Therefore, more theoretical insights are expected by the ICLR community.

I therefore recommend borderline acceptance, and encourage the authors to substantially expand the experimental scope and deepen the analysis in a future submission.

**Reviewer Concerns:**

As an evaluation-style paper, this work lacks a sufficiently broad experimental scope in terms of both datasets target model coverage.

**Reviewer Scores:**

All four reviewers will keep their original scores as 4, 8, 4, 6.

---

### Decision · Program_Chairs · 2026-01-26

Accept (Poster)